# Stable water isotope monitoring network of different water bodies in Shiyang River Basin, a typical arid river in China

Guofeng Zhu[a,b,*], Yuwei Liu[a,b], Peiji Shi[a,b], Wenxiong Jia[a,b], Junju Zhou[a,b], Yuanfeng Liu[a,b],

Xinggang Ma[a,b], Hanxiong Pan[a,b], Yu Zhang[a,b], Zhiyuan Zhang[a,b], Zhigang Sun[a,b], Leilei Yong [a,b],

Kailiang Zhao[a,b]

[a] *College of Geography and Environment Science, Northwest Normal University, Lanzhou 730070, Gansu, China*

[b] *Shiyang River Ecological Environment Observation Station, Northwest Normal University, Lanzhou 730070,*

*Gansu, China*

*Correspondence to*: *zhugf@nwnu.edu.cn*

**Abstract:** We have established a stable water isotope monitoring network in the Shiyang River

Basin in China'arid northwest. The basin is characterized by low precipitation, high evaporation

and dense population. It is the basin with the most significant ecological pressure and the greatest

water resources shortage in China. The monitoring station covers the upper, middle and lower

reaches of the river basin, with six observation systems: river source area, oasis area, reservoir

canal system area, oasis farmland area, ecological restoration area, and salinized area. All data in

the data set are differentiated by water body types (precipitation, river water, lake water,

groundwater, soil water, plant water). The data set is updated annually to gradually improve each



observation system and increase data from observation points. So far, the data have been obtained
for five consecutive years. The data set includes stable isotope data, meteorological data and
hydrological data in the Shiyang River Basin. The data set can analyze the relationship between
different water bodies and water circulation in the Shiyang River Basin. This observation
network's construction provides us with stable water isotopes data and hydrometeorological data,
and we can use theae data for hydrological and meteorological related scientific research. It can
also provide a scientific basis for water resources utilization, water conservancy project
construction, and ecological environment restoration decision-making in China's arid areas. The
data that support the findings of this study are openly available in Zhu (2021) at "Data sets of
Stable water isotope monitoring network of different water bodies in Shiyang River Basin, a
typical arid river in China (Supplemental Edition)", Mendeley Data, V1, doi:
10.17632/w5rpxwf99g.1.
**Keywords**: Stable isotopes; Arid river; Monitoring network
**1 Introduction**

Hydrogen and oxygen isotopes are useful tracers in the water cycle (Zannoni et al., 2019).

While the proportion of stable isotopes such as $\delta^2H$ and $\delta^{18}O$ in natural water bodies is small, $\delta^2H$
and $\delta^{18}O$ respond very quickly to environmental changes and historical record information on the

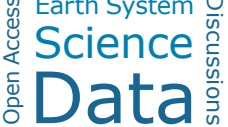

water cycle evolution. Simultaneously, the fractionation of isotopes also runs through every link of
the water cycle (Song et al.,2007; Dansgaard W, 1953;Dansgaard W, 1964). Stable isotopes of
hydrogen and oxygen in water have been widely used in the water cycle (Edwards et al., 2010;
Penna et al., 2013; Timsic et al., 2014; Evaristo et al., 2015; Negrel et al., 2016), paleoclimate and
paleoenvironmental evolution (Wei et al., 1994; Speelman et al., 2010; Steinman et al., 2010;
Hepp et al. 2015), reconstruction of pale plateau height (Thompson et al., 2000; Yao et al., 2008;
Xu et al., 2015; Li et al., 2017) and other fields. Stable isotopes provide an effective method for
studying of regional and global water cycles (Craig, 1961; Tian et al., 2001; Vallet-Coulomb et al.,
2008; Bowen et al., 2012; Gibson et al., 2017). In the water cycle, the composition of hydrogen
and oxygen isotopes in different water bodies is affected by isotope fractionation. The isotopes are
widely distributed in time and space, and different water bodies have different isotope
characteristics (Zhang et al., 2015; Christophe et al., 1998;). Precipitation stable isotopes are
affected by climate change caused by large scale weather events and local meteorological
elements, and geographical conditions. With the change of precipitation isotopes, the isotopes of
surface water and groundwater will also change in time and space sensitively (Yin et al., 2010).
Many researchers have studied stable isotopes of hydrogen and oxygen in different regions of the
world and have achieved fruitful results (Matthew et al., 2010). There are about 1400 precipitation
stable isotopes monitoring stations worldwide (IAEA/WMO, 2014). In addition to GNIP, some
national scale isotope monitoring networks have been built successively, such as Canada (Birks et
al., 2009), The United States (Vachon et al., 2007), Austria (Kralik et al., 2004), France (Chery et
al., 2004), and India (Kumar et al., 2010). Since the beginning of the 21st century, international
collaborative research programs on isotopes have been carried out with the auspices of
international organizations such as the International Atomic Energy Agency (IAWA), UNESCO
and WMO. For example, the Global Network of Isotopes in Rivers, GNIR (for short), The Isotope
Global Observation Network of Water and Carbon cycle Dynamics (LEAFLET), and the National
Coordinated Research Project (CRP) for determining farmland water cycle fluxes by applying
environmental    isotope    technology.    Compared    with    traditional    hydrology    methods,
hydrogen-oxygen stable isotope technology has significant advantages in solving the problems
such as the recharge relationship between different water bodies, soil, and plant water sources, and
the calculation of lake surface evaporation (Liu et al., 2009; Tian et al., 2009; Pu et al., 2013;
Wang et al., 2014; Wang, 2016; Ding et al., 2018). In particular, meteoric water - surface water -
soil water - groundwater can be regarded as a unified "system" to quantitatively study the
hydraulic connections between different water bodies (Burns, 2002). With the continuous
improvement of stable isotope theory and analysis and determination technology, isotope



hydrology has gradually become one of the crucial branches of hydrology. Its scope and depth of
research have also been expanded constantly. However, due to the limitations of sampling time
and space and the limitations of experimental analysis, there has always been a lack of
comprehensive research on different water bodies in the same area or basin over a long period
time, which makes it challenging to use stable isotope comparison to study the water cycle in a
specific area.

This paper compiles the stable water isotope data of the Shiyang River Basin from 2015 to

2019 and its corresponding meteorological and hydrological data into a data set. The stable
isotope data are all obtained by field sample collection and laboratory test and analysis.
Meteorological and hydrological data are obtained by weather and hydrological stations in the
Shiyang River Basin. We can ues these data to analyze the relationship between different water
bodies, understand the Shiyang River Basin water cycle process, and provide a scientific basis for
water resources utilization, water conservancy project construction, and ecological environment
decision in the arid region of China. Thus, the present study underlines the effective use of stable
isotopes in studying the hydrologic cycle, which is not yet been utilized in many parts of the world.
The Shiyang River Basin study should reference for subsequent research in arid regions within
China and other regions of the world.



## 2 Study area

The Shiyang River Basin is located in the eastern section of Qilian Mountain and Hexi Corridor, and it is the third-largest river in the flowing water system in the Hexi Corridor of Gansu Province. The topography of the Shiyang River Basin slopes sharply from southwest to northeast, with Qilian Mountains in the south, alluvial plains and Gobi in the middle, and flood plains and deserts in the north (Zhu et al., 2020). The river is about 250 km long, and the basin area is $4.16\times 10^4 km^2$ (101°41'-104°16'E and 36°29'-39°27'N). The average annual runoff is about $15.75\times 10^8 km^3$. From south to north, the Shiyang River Basin covers three different climatic regions: the southern Qilian Mountain area has an alpine and semi-arid climate (2000-5000m above sea level), with an annual average temperature below 6℃ and rainfall of 300-600mm; the central corridor plain has a dry climate (1500-2000m above sea level), the annual average temperature is between 6-8℃, and the rainfall is 150-300mm. In the north, there is an arid climate (1300-1500m above sea level), with an annual average temperature higher than 8℃ and rainfall less than 150mm (Wen et al., 2013). The precipitation in Shiyang River Basin is mainly from July to September, and the average relative humidity in summer and autumn is higher than that in winter and spring. Because the evaporation is far more than the precipitation, the farmland irrigation water in Shiyang River

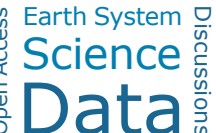

Basin is mainly surface and groundwater. The Shiyang River Basin has complete surface water
and groundwater irrigation system, irrigating 4.6 million mu of cultivated land in the basin.
**3 Observation network design**
**3.1 Site Selection**
To form a stable isotope monitoring network for different water bodies, we have set up 53
monitoring points in the Shiyang River Basin from 2015 to 2019, among which 34 were upstream,
16 in the midstream, and 3 in the downstream. Systematic sampling is carried out once a month,
and 6,760 samples have been obtained, including 1,210 precipitation samples, 1,101 surface water
samples, 161 groundwater samples, 3,779 soil water samples, and 509 plant water samples. Fig. 1
shows the distribution of stable water isotopes monitoring points in the Shiyang River Basin.
These monitoring points are located in different positions (upstream, middle and downstream)
within the basin, including six observation systems (Fig. 1): river source area, oasis area, reservoir
system area, oasis farmland area, ecological restoration area, and salinized area, which is
convenient to comprehensively analyze the microscopic water circulation process in the arid area.

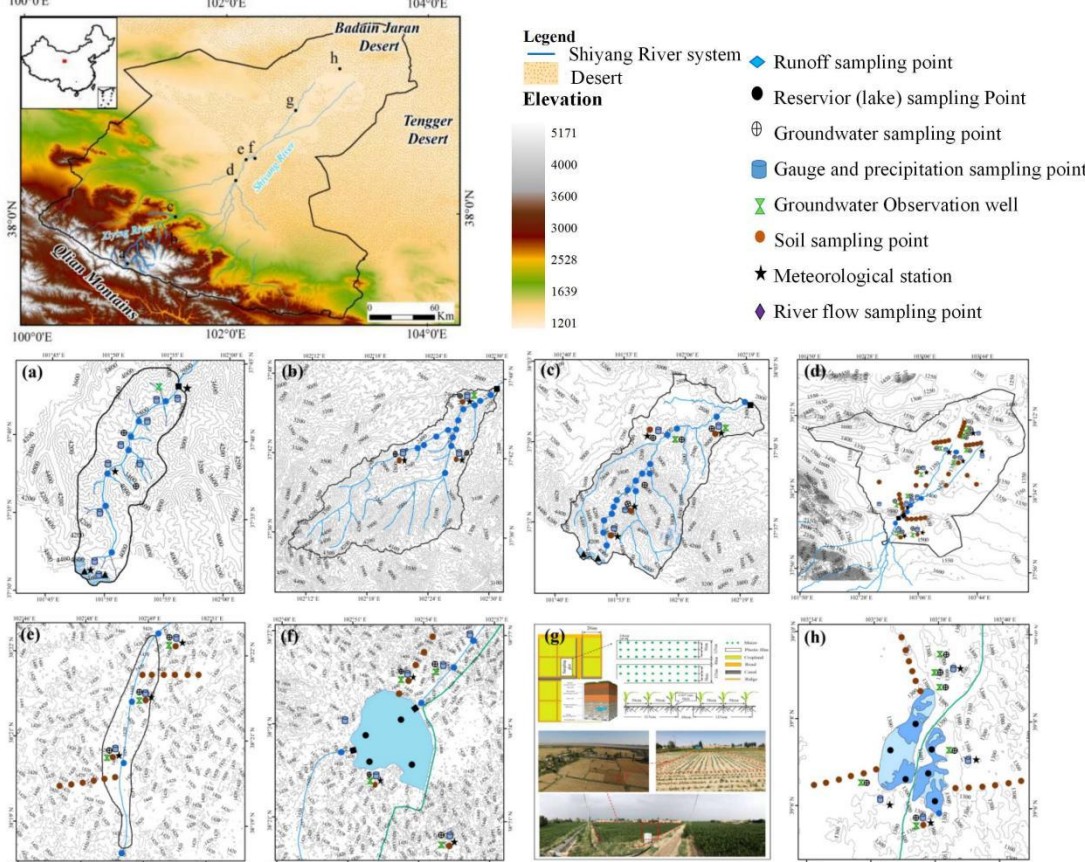


**Figure. 1 Shiyang River Basin Monitoring Network (a: Ningchang River observation system,**

**river source area; b: Ice trench observation system, river source area; c: Xiying River Basin,**

**source observation system; d: Minqin soil system, ecological restoration; e: Dongtan**

**Wetland Observation System in the middle reaches of Shiyang River, ecological restoration;**

**f: Hongyashan reservoir canal observation system, ecological restoration; g: Datan**

**Farmland Observation System, ecological restoration; h: Qingtu Lake observation system,**



**ecological restoration)**
**4 Instrument and data acquisition**
**4.1 Collection of precipitation**
In order to collect precipitation, 16 weather stations were set up in Shiyang River Basin,
which included rainfall barrels for precipitation observation and sampling. The rainfall barrels
are placed in an open place outside and composed of rain carriers, funnel, water storage bottles,
and rain cups. The diameter of the rain carrier is 20 cm, and the port of the device is horizontal.
The height of the rain-bearing mouth of the instrument is set as 70 cm from the ground plane. The
rain gauge is used to observe precipitation and collect precipitation samples. The collected liquid
precipitation is transferred to a 100 ml high-density sample bottle immediately after each
precipitation event. Measuring cylinder for solid-state precipitation, with rain collection back to
indoor at room temperature (23℃), then transferred to the high density in the sample bottle. The
sample bottles were sealed with parafilm until the end of cryopreservation, at the same time, in
samples of the polyethene bottle label, label date, type of precipitation (rain, snow, hail) and
rainfall. For the occurrence of multiple precipitation events within a day, multiple sampling is
required.
**4.2 Collection and storage of surface water and groundwater**



Polyethene bottles are used to collect surface water (rivers, lakes, reservoirs) and
groundwater samples. When collecting water samples, stratified sampling is carried out at
different depths (surface layer, middle layer, bottom layer). The bottle of the sample is sealed with
parafilm film and then frozen until the experiment. Meanwhile, a label is pasted on the polyethene
sample bottle, telling the date, sampling point, sampling depth of the sample and the stream and
tributary stream. The collected water samples should be placed in places where the sunlight is not
direct so as to avoid evaporation of water, which would affect the validity of the data.The samples
were taken back to the refrigerator in the laboratory within 10 hours.
4.3 Collection and storage of soil and plant water
The soil sample is collected at a depth of 100cm, and samples are taken sequentially at 10cm
intervals. The soil samples collected were divided into two parts, one part of which was put into a
50 ml glass bottle. The bottle mouth was sealed with parafilm membrane and transported to the
observation station within 10 hours after the sampling date was marked for cryopreservation to
test stable isotope data. The other part of the sample was placed in a 50 ml aluminium box and by
using the drying method to test the soil moisture content. Plants sample collection: sampling
scissors collected the xylem stem of vegetation, the bark was stripped and put into a 50 ml glass
bottle, sealed, and frozen until the experimental analysis.





**5 Data set**

The stable isotope data is obtained through experimental analysis, and the meteorological

data is obtained from the weather station in the Shiyang River Basin.
**5.1 Observation point**

From 2015 to 2019, a total of 53 monitoring points have been set up in the Shiyang River

Basin. For the convenience of data recording, each monitoring point is recorded in short form.
Table 1 lists each station's complete names and corresponding meteorological parameters, easy to
understand and use.
**Table 1 List of site parameters**

| Abbreviation | Full name | Longitude | Latitude | Elevation (m) | Temperature (℃) | Precipitation (mm) | Sampling type | Location |
|---|---|---|---|---|---|---|---|---|
| QHLYXM | Qinghai Forestry Project | 101°51' | 37°32' | 3899 | - | - | river water | a |
| MK | Colliery | 101°51' | 37°33' | 3647 | -0.23 | 1039.17 | precipitation | a |
| LXWL | Winding Road | 101°50' | 37°34' | 3305 | - | - | river water | a |
| SDHHC | Tunnel Junction | 101°50' | 37°34' | 3448 | - | - | river water | a |
| BDZ | Transformer Substation | 101°51' | 37°33' | 3637 | - | - | soil, plant, river water | a |
| NQ | Ningqian | 101°49 | 37°37' | 3235 | - | - | river water | a |
| SCG | Ningtanhe Middle East branch mixed | 101°50' | 37°38' | 3068 | - | - | river water, precipitation, soil | a |



| | | | | | | | |
|---|---|---|---|---|---|---|---|
| | water | | | | | | |
| MTQ | Wood Bridge | 101°53' | 37°41' | 2741 | - | - | river water | a |
| SCLK | Three-way Intersection | 101°55' | 37°43' | 2590 | - | - | river water | a |
| JTL | Nine Ridge | 102°02' | 37°51' | 2267 | - | - | groundwater | a |
| WGQ | The Bridge of the Cultural Revolution | 102°07' | 37°53' | 2174 | - | - | river water | a |
| XYSK | Xiying Reservoir | 102°12' | 37°54' | 2058 | - | - | river water | c |
| XYZ | Xiying Town | 102°26' | 37°58' | 1748 | 10.44 | 491.35 | precipitation | c |
| GGKFQ | Reform and Opening Bridge | 101°58' | 37°46' | 2590 | - | - | river water | c |
| HJX | Huajian Township | 102°00' | 37°50' | 2390 | 7.65 | 262.64 | river water, groundwater, precipitation, soil | c |
| WW | Wuwei | 102°37' | 37°53' | 1581 | 5.23 | 300.14 | river water | c |
| HLZ | Ranger Stations | 101°53' | 37°41' | 2721 | 3.25 | 469.44 | river water, precipitation, soil, plant, groundwater | a |
| LLL | Lenglong Ridge | 101°28' | 37°41' | 3500 | 5.78 | 350.34 | precipitation | a |
| ZZXL | Zhuaxi Xiulong | 103°20' | 37°18' | 3556 | -2.37 | 500.17 | precipitation | d |
| JDT | Jiudun Beach | 102°45' | 38°07' | 1464 | 10.54 | - | precipitation | d |
| SCG | Shangchigou | 102°25' | 38°03' | 2400 | 7.28 | 377.13 | precipitation, river water, groundwater | d |
| WWPD | Wuwei Basin | 102°42' | 38°06' | 1467 | - | - | precipitation, groundwater, soil, plant | d |



| DT | Dongtan | 102°47' | 38°16' | 1434 | 8.90 | 240.05 | river water, soil, plant | e |
| HYSSK | Hongyashan Reservior | 102°53' | 38°24' | 1416 | 7.81 | 100.17 | river water, groundwater, soil | f |
| CQQ | Caiqi Bridge | 102°45' | 38°13' | 1443 | 5.63 | 300.26 | groundwater, river water, soil, plant | d |
| XJG | Xiajiangou | 102°42' | 38°07' | 1200 | 9.36 | 110.18 | groundwater | d |
| HGG | Hongqi Valley | 102°50' | 38°21' | 1421 | 8.34 | 113.16 | precipitation, groundwater | d |
| BHZ | Protection Station | 102°29' | 38°09' | 2787 | - | - | groundwater | d |
| BDC | Beidong Township | 103°02' | 38°32' | 1367 | 9.52 | 155.45 | groundwater | g |
| XXWGZ | Xiyin Wugou Township | 102°58' | 38°29' | 1393 | - | - | groundwater | d |
| MQBQ | Minqin Dam | 103°08' | 39°02' | 1400 | 8.33 | 113.19 | soil | d |
| QTH | Qingtu Lake | 103°36' | 39°03' | 1313 | 7.86 | 110.79 | precipitation, groundwater, lake water, soil | h |
| SWX | Suwu Township | 103°05' | 38°36' | 1372 | 9.82 | 155.84 | groundwater, soil, plant, river water | d |
| DTX | Datan Township | 103°14' | 38°46' | 1349 | 11.49 | - | precipitation, groundwater, soi, plant, river water | g |
| YXB | Yangxia Dam | 102°41' | 38°01' | 1489 | 10.76 | - | precipitation, groundwater, soil, plant | d |
| XBZ | Xuebai Toen | 103°01 | 38°32' | 1387 | 10.77 | - | precipitation | b |
| SYQ | Laboratory | 102°22' | 37°42' | 2438 | - | - | river water, | b |




| | Area | | | | | | soil | |
|---|---|---|---|---|---|---|---|---|
| XCL | Small Valley | 102°24' | 37°43' | 2267 | - | - | river water | b |
| JCLK | Intersection | 102°20' | 37°41' | 2544 | - | - | river water, soil | b |
| QSHSY | Spring River | 102°22' | 37°38' | 2747 | - | - | spring water | b |
| HLD | Confluence | 102°26' | 37°44 | 2146 | - | - | river water, soil, plant | b |
| QXZ | Meteorological Station | 102°20' | 37°42 | 2543 | 3.34 | 510.56 | precipitation, groundwater | b |
| BGH | Binggou River | 102°17' | 37°40' | 2872 | 5.28 | - | river water, soil water, | b |
| NCHHLH | South Nancha River | 102°26' | 37°43' | 2163 | - | - | river water | b |
| LKS | Two Pine | 102°17' | 37°40' | 2832 | 5.69 | - | river water, soil | b |
| NYSKRK | Nanying Reservoir | 102°29' | 37°47' | 1955 | 7.82 | 330.16 | river water | b |
| SGZZ | Sigou stckade | 102°23' | 37°40' | 2492 | 10.34 | 675.54 | river water | b |
| JZGD | Construction Site | 102°25' | 37°41' | 2303 | - | - | river water | b |
| QLX | Qilian Township | 102°42' | 38°08' | 3394 | 5.13 | 300.15 | precipitation, spring water | d |
| XYWG | Xiying Wugou | 102°10' | 37°53' | 2097 | 7.99 | 197.67 | river water, precipitation, soil, plant | c |
| HSH | Hongshui River | 102°45' | 38°13' | 1454' | - | - | river water | d |
| XCL | Small village | 102°24' | 37°43' | 2267 | - | - | river water | b |
| YHRJ | A family | 102°20' | 37°42' | 2543 | - | - | river water | b |

**5.2 Meteorological and hydrological data set**
We obtained the meteorological data, including temperature, precipitation, atmospheric



pressure, relative humidity, wind speed, sunshine duration. Store the obtained weather data in the
corresponding weather station file. Through the classification and sorting of meteorological data,
the daily meteorological data, monthly meteorological data, seasonal meteorological data, and
annual meteorological data are formed. Finally, the meteorological data set is formed. The
obtained hydrological data includes the precipitation and flow data of each hydrological
observation point. Through the classification and arrangement of hydrological data, daily
hydrological data, monthly hydrological data, seasonal hydrological data, and annual hydrological
data are formed. Finally, the hydrological data set is formed.
**5.3 Stable water isotope data set**

The stable water isotope data set is compiled from Fig. 2. Firstly, field sampling is conducted

to obtain samples of different water bodies. The sampling interval is one month, and the data set is
updated once a year. According to the types of samples, the samples can be divided into two
categories: precipitation, river water, lake water, and groundwater can be directly tested after
filtration, while soil water and vegetation water need to be vacuum condensed and extracted to
separate the water in soil and vegetation for testing and analysis. The assembly of the data set
relies mainly on the monitoring data and instrument test data. The extraction apparatus's use is
BJJL - 2200 fully automatic vacuum condensate extraction system. Analysis instrument is LWIA -


24 d liquid water isotope analyzer. Therefore, higher requirements are put forward for the quality
and feasibility of the data. We use manner-Kendall software to test the data obtained from
meteorological and hydrological stations. The inspection of data is an important step to judge the
validity of data. The stable isotope data set and the meteorological and hydrological data set are
combined into one data set.

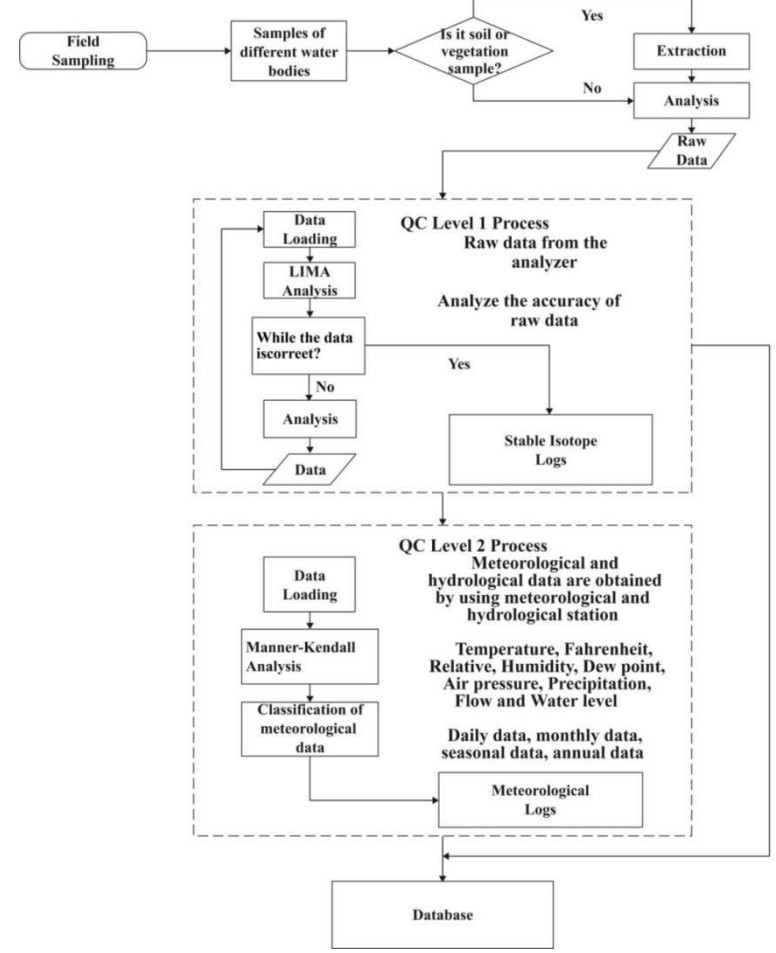


**Figure. 2 Extraction, analysis of the instrument and data set production process**
**6 Data quality**
This monitoring network aims to provide data for the Shiyang River Basin, and there can be
no great lag. In practice, some quality problems have little impact on data users, because we will
test the quality data before opening data, on the one hand, for meteorological and hydrological
data, we will use manner-Kendall software to test the isotopes data. For isotopic data, we will use
LIMA post-analysis software to select the wrong samples and reanalyze them. On the other hand,
we will screen the experimental data again and let the data's users get the quality data. At present,
the leading cause of data error is instrument error. Here are some common problems.
**6.1 Sample collection and storage**
After sample collection, extraction and analysis are needed. The extraction work is aimed at
soil and vegetation samples, which need to be stored in the freezer until the experimental analysis.
From the point of view of the collection, vegetation samples should be collected quickly.
Otherwise, resulting in a small amount of water in the vegetation, which will affect the quality of
the data. From the sample storage perspective, when too many vegetation samples are collected,
the time from sampling to extraction to analysis will be too long, and the isotope fraction caused
by evaporation will affect the test results.





**6.2 Experiments**

The experimental equipment has impurities in the pipeline, methanol, ethanol pollution and other problems, leading to errors in the experimental data. Therefore, the instrument should be checked and cleaned on time during sample analysis. After completing the experiment, we should test the the data promptly manner, and select the wrong samples.

**6.3 Modification of plant water isotope data**

Suppose the water sample contains compounds with the same absorption characteristics of the same wavelength. In that case, it will lead to errors in the measurement of the laser liquid water analyzer, and the most likely pollutants to cause errors are methanol and ethanol. So using deionized water with different concentrations of pure methanol and ethanol, the combination of Los Gatos company LWIA - Spectral Contamination Identifier v1.0 Spectral analysis software (NB) to determine methanol and ethanol (BB) pollution degree of spectrum measurement, establishing the $\delta^2H$ and $\delta^{18}O$ correction method for the spectra of pollution (Meng et al., 2012; Liu et al., 2013). In the correction process, the configuration of methanol and ethanol solution concentration was similar to Meng's experiment (2012). Correction results for methanol its broadband measurements of NB metric logarithmic respectively with $\Delta\delta^2H$ and $\Delta\delta^{18}O$ are significantly quadratic curve relationship, respectively is:



$$\Delta\delta^2\text{H} = 0.018(\ln NB)^3 + 0.092(\ln NB)^2 + 0.388\ln NB + 0.785 (R^2 = 0.991, \quad p > 0.0001) \quad (2\text{-}1)$$


$$\Delta\delta^2\text{O} = 0.017(\ln NB)^3 - 0.017(\ln NB)^2 + 0.545\ln NB + 1.356 (R^2 = 0.998, \quad p < 0.0001) \quad (2\text{-}2)$$

Its broadband measurements for ethanol correction results in BB metric and $\Delta\delta^2$H and $\Delta\delta^{18}$O, a
quadratic curve and linear relationship respectively, are:

$$\Delta\delta^2\text{H} = -85.67 BB + 93.664 (R^2 = 0.747, p = 0.026)(BB < 1.2) \quad (2\text{-}3)$$


$$\Delta\delta^2\text{O} = -21.421 BB^2 + 39.935 BB - 19.089 (R^2 = 0.769, \quad p < 0.012) \quad (2\text{-}4)$$

**7 Results and discussion**
**7.1 Stable isotopes characteristics of different water bodies**

In the catchment dominated by precipitation, the seasonal difference between $\delta^{18}$O and $\delta$D

values is large (Dansgaard W, 1964). In Fig. 3, we can be found that (1) $\delta^{18}$O and $\delta$D are periodic
with time, that is, they are depleted in winter and spring, enriched in summer and autumn, and the
value of stable isotopes reaches a high value in summer and a second high value in autumn. The
former is related to precipitation dilution, while the latter is related to high temperature and intense



evaporation. (2) $\delta^{18}O$ and $\delta D$ of lake water fluctuate more than river water, groundwater, soil
water, and plant water, because the lake's evaporation is much more robust in summer than in
other seasons. (3) The change trend of $\delta^{18}O$ and $\delta D$ in surface water is the same, the change of
groundwater lags behind that of surface water, and its change range is smaller. (4) The variation of
$\delta^{18}O$ and $\delta D$ in different water bodies are generally consistent, showing good consistency.



**Figure. 3 Distribution of different water bodies' δ¹⁸O and δD in the Shiyang River Basin**

**from 2015 to 2019**

**7.2 Changes in runoff**



According to the four hydrological observation stations in the Shiyang River Basin, the
multi-year average water level in the Shiyang River Basin from 2015 to 2019 was 9.71m. among
which the average annual water level of 2015, 2016, 2017, 2018, and 2019 are 9.56m, 10.67m,
10.11m, 7.18m, and 11.06m, respectively. In 2018, Shiyang River Basin had the lowest water level
of 7.18m. The water level in this basin peaks in summer and reaches a second peak in spring, and
the water level in Shiyang River Basin is in the rainy season in summer with more precipitation.
Spring mountain snow and ice melt supply Shiyang River related.
The annual flow of the Shiyang River Basin from 2015 to the 2019 year is 1436.04m$^3$/s,
among which the annual flow in 2015, 2016, 2017, 2018, and 2019 are 1435.9m$^3$/s, 1435.81m$^3$/s,
1436.05m$^3$/s, 1436.14m$^3$/s, and 1436.29m$^3$/s, respectively. The flow in spring and summer is
larger than that in winter and autumn. Take the year 2015 as an example, the maximum flow of the
Shiyang River Basin was 57.0m$^3$/s, which appeared on July 5. The annual runoff was 3.016 ×
10$^8$m/s, the runoff modulus was 0.936 × 10$^{-3}$m$^3$/(S.km$^2$), and the runoff depth was 29.5mm. The
largest flood volume1day, 3days, 7days, 15days, 30days, and 60days occurred on July 5, July 4,
July 4, July 2, July 2, and June 22.

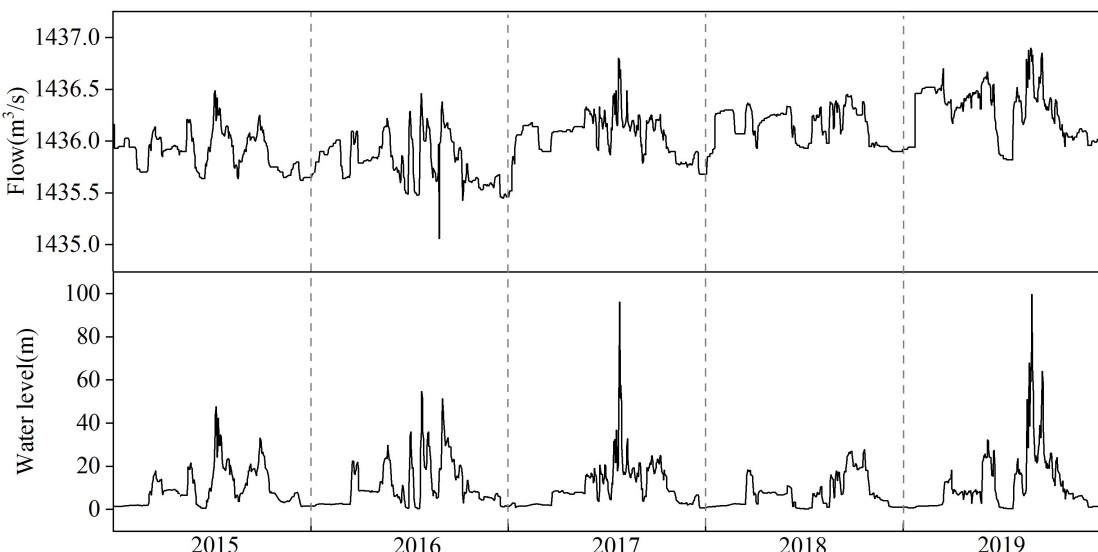


**Figure. 4 Changes of hydrological data of the Shiyang River Basin from 2015 to 2019**


**7.3 Connections between different bodies of water**


Based on the precipitation isotope data of the Shiyang River Basin from January 2016 to
December 2019 (Fig. 5), using the least squares of the LMWL of the Shiyang River Basin, the
local waterline equation (LMWL) is obtained: $\delta D = 7.65\delta^{18}O + 9.75$, compared to the global
atmosphere waterline equation (GMWL), slope and intercept are small, but $\delta D$ and $\delta^{18}O$ maintain
a good linear relationship ($R^2 = 0.96$), which is related to the geographical location of the study
area. The Shiyang River Basin is located in the northwest inland of China, and the climate
environment is dry. It is subject to intense secondary evaporation during the precipitation, making
the slope and intercepts relatively small. It also reflects the existence of stable isotope unbalanced



fractionation effect under the arid climate background.
Precipitation, river water, lake water, groundwater, soil water, and plant water are distributed
near GMWL, indicating that they share the same water source. The deviation of the lake from
GMWL indicates that it experienced intense evaporation.
By comparing the slope and intercept of the relation expressions $\delta^{18}O$ and $\delta D$ of GMWL and
different water bodies, it can be seen that, as far as the slope is concerned, precipitation is the
highest (7.65), followed by groundwater (5.11), lake water is the lowest (2.14). There is little
difference between the slope of precipitation and groundwater, which means there is a mutual
recharge relationship. In terms of intercept (d), the precipitation was the highest (9.75), followed
by the river (-8.44). When the water body evaporates in the unsaturated atmosphere, the light
isotopes evaporate preferentially. The combined effect of the dynamic fractionation effect of the
river accelerates the ratio of the $\delta D$ and $\delta^{18}O$ fractionation effects in the evaporated water vapor,
resulting in an increase in d in the water vapor and a decrease in d in the remaining water body.
The average value of $\delta^{18}O$ and $\delta D$ of soil water is between plant water and precipitation, but closer
to precipitation (Table 2), indicating that the soil is mainly recharged by precipitation. In the $\delta^{18}O$
and $\delta D$ equations of precipitation, lake water, soil water, river water, plant water, and groundwater,
$R^2$ decreases in turn, and the linear relationship between $\delta^{18}O$ and $\delta D$ becomes smaller and smaller.

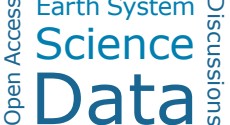

These phenomena indicate that different water bodies have different degrees of mutual
complementarity. Among them, soil water is the most miscible and is supplied by multiple water
sources.

Take $\delta^{18}O$ for example, in terms of the variation coefficient, the absolute value of stable

isotopes (4.4) of the lake water is far higher than that of the other five water bodies (groundwater,
river water, soil water, precipitation, plant water: 0.08, 0.11, 0.37, 0.71, 2.54), reflecting the high
volatility of the lake water.

The correlation coefficient between $\delta^{18}O$ and $\delta D$ of lake water, groundwater, and plant water

is relatively low. The evaporation of lake water in summer is particularly intense, which leads to
the great difference in winter and summer. The stable isotopic value of lake water varies
significantly in different seasons, leading to a small correlation coefficient between them. The
main recharge source of groundwater and plant water is meteoric water. It takes a certain time for
meteoric water to converge into surface water and groundwater, leading to isotopic fraction,
leading to a small correlation coefficient between $\delta^{18}O$ and $\delta D$ of the two water bodies.





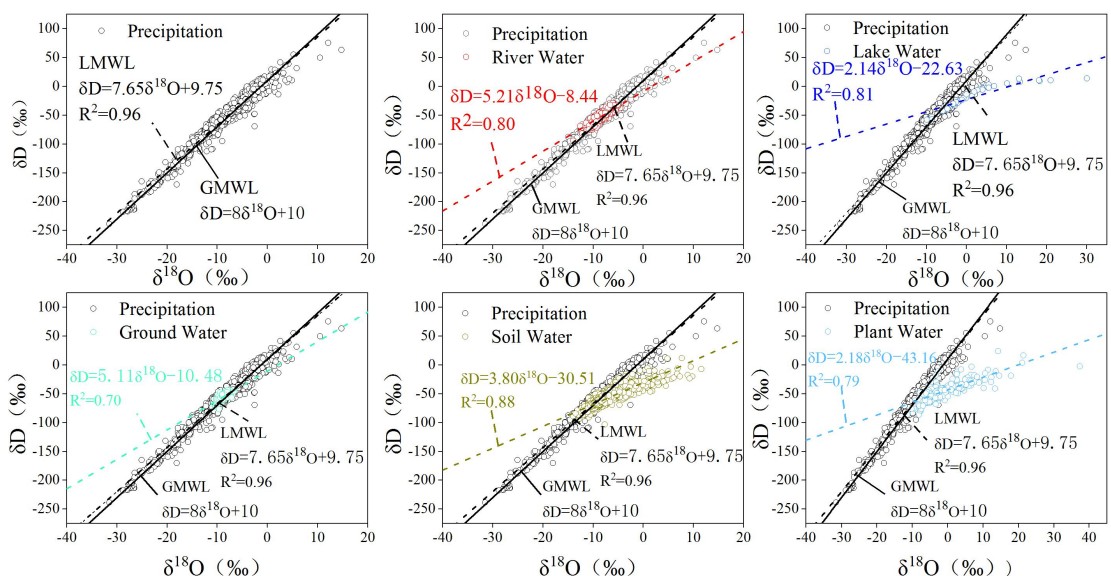

Figure. 5 The change of δ¹⁸O and δD in different water bodies in the Shiyang River Basin

Table 2 Comparison of water bodies δ¹⁸O and δD in the Shiyang River Basin from 2015 to

2019

| Water Type | δD(‰) | | | | δ¹⁸O(‰) | | | |
|---|---|---|---|---|---|---|---|---|
| | Min | Max | Average | Coefficient of variation | Min | Max | Average | Coefficient of variation |
| Precipitation | -238.62 | 75.41 | -54.63 | -0.85 | -31.22 | 14.79 | -8.39 | -0.71 |
| River Water | -94.14 | -28.89 | -53.37 | -0.12 | -13.98 | -3.44 | -8.62 | -0.11 |
| Lake Water | -57.84 | 13.56 | -18.43 | -1.11 | -9.86 | 30.01 | 1.96 | 4.4 |
| Underground Water | -76.99 | -43.72 | -52.42 | -0.10 | -10.44 | -6.57 | -8.80 | -0.08 |
| Soil Water | -102.95 | 11.81 | -59.39 | -0.20 | -13.94 | 11.62 | -7.61 | -0.37 |
| Plant Water | -86.41 | 23.87 | -48.15 | -0.32 | -11.43 | 37.37 | -2.27 | -2.54 |

## 8 Data availability

The data that support the findings of this study are openly available in Zhu (2021) at "Data sets



of Stable water isotope monitoring network of different water bodies in Shiyang River Basin, a
typical arid river in China (Supplemental Edition)", Mendeley Data, V1, doi:
10.17632/w5rpxwf99g.1.
**9 Summary and outlook**
The data set provides a new observation and data basis for studying stable water isotopes in
different water bodies in China's inland river basins. Through these data, we can compare the
stable isotopes characteristics of different water bodies and study the correlation between different
water bodies, thus providing some guidance for the rational use of water resources in arid regions.
The data set will be updated year by year as observations are made. To improve this data set, we
encourage data set users to contact the author with suggestions.
**Author contributions**
Guofeng Zhu, and Yuwei Liu conceived the idea of the study; Peiji Shi, Wenxiong Jia and
Junju Zhou set up observation system; Xinggang Ma, Hanxiong Pan,Yu, Zhang, Zhiyuan Zhang
and Leilei Yong were responsible for field sampling; Zhigang Sun participated in the experiment;
Kailiang Zhao and Yuanfeng Liu participated in the drawing; Yuwei Liu wrote the paper; All
authors discussed the results and revised the manuscript.
**Competing interests**



The authors declare no competing interests.

**Acknowledgement**
This research was financially supported by China's National Natural Science Foundation
(41867030, 41971036, 41661005). The authors thank the colleagues in the Northwest Normal
University for their help in fieldwork, laboratory analysis, and data processing.

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
