# Peer review of "water bodies in Shiyang River Basin, a typical arid"

_Earth System Science Data, 2021_

## Referee Comment (RC2)

In the previous 30 years, I have done a lot of similar work in the European Alps and Asian Hengduan Mountains, but I have to say that this is an impressive article. The author measured the stable isotopes of different water bodies in the whole Shiyang River Basin and collected corresponding hydrological and meteorological data in this manuscript. The data set from 2015 to 2019, with 53 observation points and 6760 experimental data obtained. This data set is one of the most systematic data sets (the matching of different water bodies is almost perfect) I have seen so far, and it is very influential, the key is that the data are all sample experimental data. I checked the data set, and the observation is very systematic and scientific (at least in the various documents I have seen ). I believe that the publication of this data set will promote the research of global isotope hydrology. Therefore, I support the publication of this article as soon as possible! However, a significant article must have good writing, and this article needs further improvement in expression and language.

**Major comments**

1. Data articles should be easy to read and use by other researchers, the entire manuscript, including the data set, lacks some basic information, especially information about experiments and sample collection. The six observation systems established by the author are excellent and should be illustrated the purpose of observation, you cannot expect that every reader is a professional, and the writing should be clear.

2. The author has been publishing data continuously, and there are many other data sets worldwide. The compatibility and matching of data should be considered.

Therefore, the author should add isotope experiments in the current manuscript, especially the reference standards for isotope data, which is very important for data quality.

3. A good author should think critically about the problem. In the current version of the manuscript, I have not seen the author's critical comments on stable isotope technology.

4. In addition to providing rich data to readers, data articles should also guide readers to use these data to solve scientific problems. In the current version of the manuscript, the author's outlook on the data set is short, and it is difficult for readers to be inspired by this article.

5. As a data set article, there are many soil and vegetation data in the data set. Among them, there are 3,779 soil samples and 509 plant samples. The acquisition of these data is essential. I believe this is also the study of agricultural activities and crop water use in the arid regions of Central Asia. However, the introductory part of the article focuses on the indications of stable isotopes of precipitation to the water cycle. It is recommended that the relevant discussions on isotope ecology be added to the introductory part of the article.

**Specific comments:**

1. L11-12: I think time information should be added here.

2. L15-16: Arrange six observation systems in the order of upstream, midstream, and downstream.

3. L21: Change stable isotope data to water stable isotope data, the same in other parts of the manuscript, please keep the terminology consistent in the manuscript.

4. L24: "these " not "theae ".

5. L26: How to provide a scientific basis for the construction of water conservancy projects in arid areas? The author did not mention in the manuscript.

6. L37: The format of the references needs to be revised.

7. L41: "Hepp et al., 2015" not "Hepp et al. 2015", please pay attention to the punctuation in the manuscript.

8. L49: I think adding the control factors of other water body isotopes here will be a good combination with the previous precipitation isotope factors.

9. L62: Compared with traditional hydrological methods, what disadvantages are hydrogen and oxygen stable isotope technology? In the current manuscript, I have not seen critical comments on the stable isotopes of hydrogen and oxygen.

10. L101:Replace the description with exact data.

11. L114-115: The purpose of each observing system should be introduced.

12. L128: Information about the device used to collect precipitation, such as pictures, should be added.

13. L132: This sentence is repeated, and it is recommended to delete it.

14. L138-139: How to calculate the precipitation isotope value of that day after

sampling multiple times of precipitation in one day?

15. L150: Are all soil samples at 10cm intervals? I saw 5cm intervals in the data set.

16. L151-152: Are there any replicates for soil samples of each soil layer?

17. L155-157: How many plant species are sampled? How about the position of sampled stems in the canopy? What is the size of stem samples? "xylem stem" should be "stem".

18. L141: Which reservoir of water was measured?

19. L142: How is the groundwater sampled? What is the depth of water table at each

20. sampling point.

21. L145: "telling the date"?

22. L146-147: Where is the water sample placed?

23. L150: What types of soil are collected?

24. L177-190: The article did not mention the accuracy of the hydrogen and oxygen stable isotope data and the standard samples used in the experiment, which is missing for an article introducing the data.

25. L197: "to test the hydrological data" not "to test the isotopes data".

26. L199: How to screen experimental data?

27. L232:Both "$\delta D$" and "$\delta^2 H$" are used in the manuscript. I suggest use one of them.

28. L233: "...we can found that...".

29. L264-266: This sentence lacks a subject.

30. L266-267: Please check the full names of LMWL and GMWL.

31. L278-281: What does the data in brackets mean?

32. L309: Both "underground weater" and "ground water" are used in the manuscript. I suggest use one of them.

Figure 1: "Shiyang River system"? Is it "Shiyang River Basin"? Improve the clarity of the picture. The picture in the current manuscript is very blurry, so I can't get relevant information from the picture well.

Figure 3: I think it is easier to compare the isotopes of different water bodies on one picture.

Table 1: The unit is unclear. It is not clear whether the precipitation is annual or multi-year average? It is also unclear whether the temperature is air temperature? It is recommended to arrange the sampling points in order from upstream to downstream.

Table 2: Please change Table 2 to a three-line table.

---

## Author Comment (AC1)

**Response report**

Dear Reviewer:

Thank you for your letter and for the reviewer's comments concerning our manuscript entitled "Stable water isotope monitoring network of different water bodies in Shiyang River basin, a typical arid river in China" (Manuscript Number: essd-2021-79).

According to the reviewer's comments, we have carefully checked the data and research methods, and seriously modify our manuscript. The modified portions have been marked in red in the manuscript changes version. The primary corrections and the response to the reviewers' comments are as follows.

**Responses to the reviewer's comments:**

**Zhu et al. present a measurement campaign for stable water isotope data measured in river water, ground water, soil water, precipitation and xylem water in the Shiyan River Basin, China. The described data sets starts in the year 2015 and ends in 2019. Principally, the data set is very impressive due to their many measurement but the paper, as well as the data set, are missing some basic information, mainly about the methods, to have a significant and useful impact to the field of isotope hydrology. I think that in the presented form the scientific community cannot use fully confident the data set due missing information about data quality, measurement uncertainty and measurement process. Given the explanation below, I think the paper does not fulfill the requirement for publication, yet. I would suggest major revisions.**

Response: We have gradually improved it by carefully revising every recommendation you mentioned.

**Major concerns**

**Regarding writing, style and readability:**

**1. Please, structure your text using paragraphs / line breaks! (especially in the introduction section)**

Response: Thank you very much for your suggestion. We have organized our article into segments according to your suggestion. The revised Introduction section is as

follows:

  Hydrogen and oxygen isotopes are useful tracers in the water cycle (Zannoni et al., 2019). While the proportion of stable isotopes such as δD and δ¹⁸O in natural water bodies is small, δD and δ¹⁸O respond very quickly to environmental changes and historical record information on the water cycle evolution (Vandenschrick et al., 2002). Simultaneously, the fractionation of isotopes also runs through every link of the water cycle (Song et al., 2007; Dansgaard W, 1953, 1964). For example, Meißner et al. (2014) emphasized that the change of δ¹⁸O largely depends on the soil type (Araguás-Araguás et al., 1995). Orlowskii et al. (2016) showed that incomplete water extraction in the cryogenic distillation process might fractionate water isotopes. Therefore, clay requires a longer extraction time and temperature to reduce the fractionation effect in the extraction process. In addition, a study by Sofer and Gat in 1975 showed that the formation of hydrated spheres around cations in aqueous solutions would fractionate the oxygen isotopes of the water. Gaj et al. (2017) showed that the isotope characteristics are biased due to a process different from Rayleigh distillation that we cannot reduce the effect caused by the mineral-water interaction entirely.

  Stable isotopes of hydrogen and oxygen in water have been widely used in the water cycle (Gibson et al., 2010; Penna et al., 2013; Timsic et al., 2014; Evaristo et al., 2015; Negrel et al., 2016), paleoclimate and paleoenvironmental evolution (Wei et al., 1994; Speelman et al., 2010; Steinman et al., 2010; Hepp et al., 2015), reconstruction of pale plateau height (Thompson et al., 2000; Yao et al., 2008; Xu et al., 2015; Li et al., 2017) and other fields. Stable isotopes provide an effective method for studying regional and global water cycles (Craig, 1961; Tian et al., 2001; Vallet-Coulomb et al., 2008; Bowen et al., 2012; Gibson et al., 2017). In the water cycle, the composition of hydrogen and oxygen isotopes in different water bodies is affected by isotope fractionation (Gat et al., 1996; Ma et al., 2012; Sun et al., 2012 ). The isotopes are widely distributed in time and space, and different water bodies have different isotope characteristics (Yang et al., 2013; Liu et al., 2019). Precipitation stable isotopes are affected by climate change caused by large-scale weather events, local meteorological

elements, and geographical conditions. With the change of precipitation isotopes, the isotopes of surface water and groundwater will also change in time and space sensitively (Yin et al., 2010).

Many researchers have studied stable isotopes of hydrogen and oxygen in different world regions and have achieved fruitful results (Risi et al., 2013; Pfahl et al., 2008). There are about 1400 precipitation stable isotopes monitoring stations worldwide (IAEA/WMO, 2014). In addition to GNIP, some national scale isotope monitoring networks have been built successively, such as Canada (Birks et al., 2009), The United States (Vachon et al., 2007), Austria (Kralik et al., 2004), France (Chery et al., 2004), and India (Kumar et al., 2010). Since the beginning of the 21st century, international collaborative research programs on isotopes have been carried out with the auspices of international organizations such as the International Atomic Energy Agency (IAWA), UNESCO and WMO. For example, the Global Network of Isotopes in Rivers, GNIR (for short), The Isotope Global Observation Network of Water and Carbon cycle Dynamics (LEAFLET), and the National Coordinated Research Project (CRP) for determining farmland water cycle fluxes by applying environmental isotope technology.

Studies have shown that physicochemical soil properties may cause the fractionation of hydrogen and oxygen in soil water (Meißner et al., 2014). Because we do not know whether the unstable hydrogen fraction during the low-temperature extraction process will cause isotope fractionation (Orlowski et al., 2016). In addition, we know little about the effect of soil microbial activity on the extracted water isotope results (Orlowski et al., 2018). However, from previous studies, it is still difficult to make post-correction in terms of soil properties or the effects of extraction conditions because such information is rarely reported, and huge variability in method details is common (Walker et al., 1994). We have always known that these "problems" exist, but water vacuum extraction is still the standard method for extracting soil and plant water in ecological hydrology (Ingraham et al., 1992). In most plants, the isotopic composition of water does not change due to root absorption and transport through the stem xylem (White et al., 1985). However, more and more studies have shown a

difference between the isotope composition of xylem water and plant water sources (Poca et al., 2019) and the fractionation can occur along the root water absorption pathway. This fact does not make the isotope method in the soil-plant-atmosphere useless to track water in the continuum, and $\delta D$ and $\delta^{18}O$ can still be used similarly to study water absorption of various plants (Poca et al., 2019). Compared with traditional hydrology methods, hydrogen-oxygen stable isotope technology has significant advantages in solving the problems such as the recharge relationship between different water bodies, soil, and plant water sources, and the calculation of lake surface evaporation (Liu et al., 2009; Tian et al., 2009; Pu et al., 2013; Wang et al., 2014; Wang, 2016; Ding et al., 2018). In particular, meteoric water - surface water - soil water - groundwater can be regarded as a unified "system" to quantitatively study the hydraulic connections between different water bodies (Burns, 2002). With the continuous improvement of stable isotope theory and analysis and determination technology, isotope hydrology has gradually become one of the crucial branches of hydrology. Its scope and depth of research have also been expanded constantly. However, due to the limitations of sampling time and space and the limitations of experimental analysis, there has always been a lack of comprehensive research on different water bodies in the same area or basin over a long time, which makes it challenging to use stable isotope comparison to study the water cycle in a specific area.

The Shiyang River Basin is one of the earliest developed river basins in Northwest China and is considered one of the most severely over-exploited inland river basins in China (Kang et al., 2004). It is one of the most prominent areas of water conflict and ecological environment problems. This paper compiles the stable water isotope data of the Shiyang River Basin from 2015 to 2019 and its corresponding meteorological and hydrological data into a data set. This work will help clarify the local water cycle in the Shiyang River Basin and the impact of human activities on agricultural production, analysis the development trend of the Shiyang River Basin under the background of global warming, provide a certain scientific reference for other semi-arid regions in the world, and promote ecological restoration

in the arid regions.

**2. The citation style and font is not consistent. A few times the first name of an author was used in the citation (see specific comments below). Additionally, the order of the bibliography is not always correct. Please, follow the journal guidelines.**

Response: Thank you very much for your suggestion. We have followed the journal guidelines to unify the citation style and font and checked and modified the references.

**3. Units are sometimes with space sometimes without space between number and unit. Please, follow the journal guidelines.**

Response: Thank you very much for your suggestion. We have unified the space between the number and unit of the article following the requirements of the journal.

**4. The structure of the chapters is not clear. Information about the methods, which ideally would be together in one section, are separated into different chapters.**

Response: Thank you very much for your suggestion. We have put the information about the method in the same chapter and the adjusted structure is as follows:

1. Introduction

2. Study area

3. Observation network design

3.1 Site selection

3.2 Observation point

4. Data and Methods

4.1 Sample collection

4.1.1 Collection of precipitation

4.1.2 Collection of surface water and groundwater

4.1.3 Collection of soil and plant water

4.1.4 Collection of meteorological data

4.2 Experiment analysis

4.2.1 Water extraction experiment

**5. Overall, I think that major parts of the paper should be rewritten.**

Response: Thank you very much for your suggestion. We have rewritten the main part of the paper, the revised content is as follows:

**4 Data and Methods**

**4.1 Sample collection**

**4.1.1 Collection of precipitation**

We have set up 16 weather stations in the Shiyang River Basin to collect precipitation, including rain barrels used to collect precipitation. The rain barrel is placed in an open outdoor area and consists of rain gear, funnel, water bottle and rain cup. The diameter of the rain gear is 20 cm, and the port of the device is horizontal. The height of the rain opening of the instrument is set to 70 cm from the ground level. A ping-pong ball is placed at the mouth of the funnel, and a layer of paraffin oil is added to the bottom of the containerto prevent isotope fractionation caused by evaporation. Immediately after each precipitation event, transfer the collected liquid precipitation to a 100 ml high-density sample bottle. Solid precipitation must be transferred to a high-density polyethene sample bottle after it becomes liquid water at

room temperature (23°C). The sample bottle is sealed with Parafilm. The polyethene bottle is labelled simultaneously, indicating the date of collecting the sample, the type of precipitation (rain, snow, hail), and the volume of precipitation. Store the collected samples in a refrigerator at about 4°C for later analysis. For multiple precipitation events in one day, various samplings are required.

**4.1.2 Collection of surface water and groundwater**

Polyethene bottles are used to collect surface water (rivers, lakes, reservoirs). Stratified sampling is carried out at different depths (surface layer, middle layer, bottom layer. Groundwater samples were obtained from the groundwater monitoring wells of the Shiyang River Basin Administration, China Hydrological Administration and Gansu Hydrological Administration. The bottle of the sample is sealed with parafilm film and then frozen until the experiment. Simultaneously, the polyethene bottle sample is labelled with the sample's date, sampling depth, and the stream and tributary stream. The collected water samples should be placed in places where the sunlight is not direct to avoid the evaporation of water, which would affect the validity of the data. The samples were taken back to the refrigerator in the laboratory within 10 hours.

**4.1.3 Collection of soil and plant water**

The soil sample is collected at a depth of 100 cm, and samples are taken sequentially at 10 cm intervals. The upper reaches of the Shiyang River Basin are mainly clay, and the middle and lower reaches are clay and sand, but sand is the main soil type. Table 2 shows the characteristics of soil in the farmland area of Minqin Oasis. The soil samples collected were divided into two parts, one of which was put into a 50 ml glass bottle. The bottle mouth was sealed with parafilm membrane and transported to the observation station within 10 hours after the sampling date was marked for cryopreservation to test stable isotope data. The other part of the sample was placed in a 50 ml aluminium box and used the drying method to test the soil moisture content since 2019.

Table 2 Basic information of soil samples (Oasis farmland area, Zhu et al., 2021)

| Soil depth (cm) | Clay (%) | Silt (%) | Sand (%) | Soil bulk density (g/cm$^3$) |
|---|---|---|---|---|
| 0-10 | 10.20 | 38.85 | 50.95 | 1.052 |
| 10-20 | 12.94 | 37.76 | 49.30 | 1.194 |
| 20-30 | 10.33 | 44.23 | 45.44 | 1.298 |
| 30-40 | 13.48 | 38.69 | 47.83 | 1.180 |
| 40-50 | 12.01 | 35.09 | 52.90 | 1.140 |
| 50-60 | 11.21 | 42.83 | 45.96 | 1.206 |
| 60-70 | 10.34 | 42.98 | 46.68 | 1.208 |
| 70-80 | 11.09 | 38.96 | 49.95 | 1.106 |
| 80-90 | 11.75 | 37.72 | 50.53 | 1.200 |
| 90-100 | 7.21 | 35.97 | 56.82 | 1.272 |

Plant sample collection: For trees and shrubs, we collect non-green branches, and for herbs, we collect non-green parts at the junction of rhizomes. When sampling, we use scissors to collect vegetation xylem stems, peel off the bark, put them in 50ml glass bottles, seal them, and freeze them until experimental analysis. Table 3 shows the plant information collected in the Shiyang River Basin

Table 3 Basic information of plant samples

| Sampling points | Vegetation types | Sample size |
|---|---|---|
| BDZ | Agropyron cristatum | 30g ±0.5 |
| CQQ | Corn (leaf), reed, jujube (Branches), dryland willow (Branches) | 30g-100g |
| DT | Reed | 30g±0.5 |
| DTX | Spring wheat (stem), corn (root, stem, leaf) | 30g±0.5 |
| HJX | Willow (Branches) | 100g±0.5 |
| HLD | Qinghai Spruce (Branches) | 100g±0.5 |
| HLZ | Qinghai Spruce (Branches) | 100g±0.5 |
| WWPD | Corn (leaf), wheat (leaf) | 30g±0.5 |
| XYWG | Poplar (Branches), wheat | 100g±0.5 |
| YXB | Corn (leaf) | 30g±0.5 |
| SWX | Corn, wheat (leaf) | 30g±0.5 |
| LLL | Salsola purpurea | 30g±0.5 |

**4.1.4 Collection of meteorological data**

The local meteorological data were obtained and recorded during the sampling period by the automatic weather stations (watchdog 2000 series weather stations) erected near the sample plot. Meteorological data include temperature (℃), relative humidity (%), atmospheric pressure (hPa), dew point temperature (℃) and precipitation amount (mm).

**4.2 Experiment analysis**

**4.2.1 Water extraction experiment**

We use vacuum condensation to extract soil and plant water. The extraction equipment used is LI-2100 automatic vacuum condensation extraction equipment. Before water extraction is performed on the soil and plants, the collected samples need to be taken out of the refrigerator to thaw, and each sample bottle should be stuffed with a small ball of cotton to prevent the water from evaporating. When extracting water, we set the extraction time to 150 minutes (180 minutes for plants), the temperature to 190°C, the upper limit of the vacuum pressure to 800pa, and the leakage rate to 0. The water evaporates from the soil or plant sample by heating it for a specified time and then freezes it in a liquid nitrogen cold trap. After the extraction is completed, the sample is thawed at room temperature and shaken. Use a 1ml syringe to extract the water sample into a labelled sample bottle, seal it and wait for the isotope experiment.

**4.2.2 Isotope experiment**

All the collected water samples were analyzed in the stable isotope laboratory of Northwest Normal University using liquid water isotope analysis (DLT-100, Los Gatos Research, USA). Each water sample and isotope standard sample were injected six times in a row. We discarded the first two injections and used the average of the last four times as the final result to eliminate the instrument memory effect. The result of the isotope measurement is expressed by the symbol "δ" and expressed in thousandths of the difference relative to the Vienna Standard Mean Ocean Water (Craig, 1961):

$$\delta_{sample}\ (‰) = \left[ \left( \frac{R_{sample}}{R_{v\text{-}smow}} \right) - 1 \right] \times 1000$$

(1)

In the formula, Rsample is the ratio of $^{18}O/^{16}O$ or $^{2}H/^{1}H$ in the collected sample, Rv-smow is the ratio of $^{18}O/^{16}O$ or $^{2}H/^{1}H$ in the Vienna standard sample.The analytical accuracy of $\delta^{2}H$ and $\delta^{18}O$ are ±0.6‰ and ±0.2‰, respectively.

4.2.3 Modification of plant water isotope data

Suppose the water sample contains compounds with the same absorption characteristics of the same wavelength. In that case, it will lead to errors in the measurement of the laser liquid water analyzer, and the most likely pollutants to cause errors are methanol and ethanol. So using deionized water with different concentrations of pure methanol and ethanol, the combination of Los Gatos company LWIA - Spectral Contamination Identifier v1.0 Spectral analysis software (NB) to determine methanol and ethanol (BB) pollution degree of spectrum measurement, establishing the δD and δ¹⁸O correction method for the spectra of pollution (Meng et al., 2012; Liu et al., 2013). In the correction process, methanol and ethanol solution concentration configuration were similar to Meng' s experiment (2012). Correction results for methanol its broadband measurements of NB metric logarithmic respectively with ΔδD and Δδ¹⁸O are significantly quadratic curve relationship, respectively is:

$$\Delta \delta D = 0.018(\ln NB)^3 + 0.092(\ln NB)^2 + 0.388\ln NB + 0.785 (R^2 = 0.991,\ p > 0.0001) \quad (2\text{-}1)$$

$$\Delta \delta^{18}O = 0.017(\ln NB)^3 - 0.017(\ln NB)^2 + 0.545\ln NB + 1.356 (R^2 = 0.998,\ p < 0.0001) \quad (2\text{-}2)$$

Its broadband measurements for ethanol correction results in BB metric and ΔδD and Δδ¹⁸O, a quadratic curve and linear relationship respectively, are:

$$\Delta \delta D = -85.67 BB + 93.664 (R^2 = 0.747, p = 0.026)(BB < 1.2) \quad (2\text{-}3)$$

$$\Delta \delta^{18}O = -21.421 BB^2 + 39.935 BB - 19.089 (R^2 = 0.769,\ p < 0.012) \quad (2\text{-}4)$$

**4.3 Data quality**

It has always been a difficult problem to control the experimental error to the minimum. We use Manner-Kendall to test meteorological and hydrological data, eliminate abnormal values, and use interpolation to obtain vacant values. For the isotope data, we first use the LIMA software to check the original isotope data. We stipulate that when one or more of the 6 data of a sample is marked in red, we call the

sample an error sample, even though the sample is in the display on the liquid water analyzer is normal, and we will re-experiment the sample until it passes the LIMA software verification. Then we will use SPSS software to check the normality of the obtained isotope data. At present, our experimental errors mainly come from the following aspects:

**4.3.1 Sample collection**

The error caused by precipitation sample collection mainly comes from the sampling personnel's failure to transfer the rainwater in the rain gauge or mixing two or more rainwaters after each precipitation event, which will cause the rainwater in the rain gauge to evaporate and make The measured value deviates greatly from the actual value.

The error caused by the collection of vegetation samples mainly comes from the slow speed of collecting samples, and the vegetation is exposed to the air for a long time, which causes the vegetation to fractionate water.

The error in collecting soil samples is that we do not know much about the influence of soil microbial activities on the extracted water isotope results. Therefore, we did not pay too much attention to the microorganisms in the soil when collecting the samples. As a result, the collected soil samples contained many microorganisms, which greatly impacted the data results.

**4.3. 2 Experiment**

The experimental error is mainly because we set the same moisture extraction parameters for samples with different soil characteristics. It is difficult to make post-mortem corrections for soil properties or the effects of extraction conditions because such information is rarely reported, and massive variability in method details is common (Walker et al., 1994). In addition, there are still measurement uncertainties during the extraction of water, which also come from the loss of water vapor during the vacuum of the extraction system and the non-temperature heating temperature, which will lead to experimental errors.

Our calibration of plant sample data only considers methanol and ethanol pollution, but the plant and soil water extracts may contain various other pollutants,

leading to experimental errors. In addition, studies have shown that the mismatch between xylem and plant water sources is due to the fractionation of isotopes in the process of water absorption (Poca et al., 2019), which questioned the fact that plants do not undergo fractionation during the process of water absorption (Porporato, 2001; Meissne et al., 2014) this traditional view. However, in the experiment, we still use traditional viewpoints as assumptions, leading to experimental errors.

**Regarding the data set available under**

**https://data.mendeley.com/datasets/w5rpxwf99g/1**

**1. The precipitation data is missing any numbers about the volume. Do we see mixed samples of the rain event?**

Response: Thank you very much for your suggestion. We have added precipitation volume information to the data set. The data set is available in Zhu, Guofeng (2021), "Stable water isotope monitoring network of different water bo dies in Shiyang River Basin, a typical arid river in China (Supplemental Edition 2021 0808)", Mendeley Data, V1, doi: 10.17632/d5kzm92nn3.1.

**2. The soil data is missing the information about the soil water content which, according to the paper, was measured.**

Response: Thank you very much for your suggestion. We have added the information of soil water content to the data set. The data set is available in Zhu, Guofeng (2021), "Stable water isotope monitoring network of different water bo dies in Shiyang River Basin, a typical arid river in China (Supplemental Edition 2021 0808)", Mendeley Data, V1, doi: 10.17632/d5kzm92nn3.1.

**3. Some data points are missing the day in the date. This is difficult to handle for any software like R to further process the data. As well, the paper does not tell anything about how to handle these measurements. Do we see monthly average values?**

Response: Yes. Since it is difficult for any software like R to deal with the missing data of a certain day in the date, we adopt the method of averaging. Therefore, in the article, we replace the data on a specific day in the missing date with the average value of the month.

**Regarding the data measurements**

**1. In my point of view, the most important information would be how the stable water isotope data were exactly measured. This includes:**

**(1)Which uncertainties can be expected due to the measurement unit?**

Response: Our stable isotope results are expressed in one-thousandths of the standard average seawater in Vienna. Because of this unit, we can expect the following uncertainties:

(1) Reliability of isotope. Since the standard sample used in our experiment is expressed in one-thousandth of the Vienna seawater, which is consistent with the internationally referenced standard, the data obtained is reliable.

(2) The accuracy of the isotope. The analytical precision of our isotope $\delta^2H$ and $\delta^{18}O$ are $\pm0.6‰$ and $\pm0.2‰$, respectively.

(3) The memory effect of the isotope. To eliminate the memory effect of the instrument, we round off the values of the first two stitches and take the average value of the last four stitches as the final result value.

**(2)How did you exactly extract the water from the xylem or the soil?**

Response: We use vacuum condensation extraction for plant water and soil water, and the instrument used is the LI-2100 automatic vacuum condensation extraction system (as shown in the figure below). The system adopts the principle of ultra-low pressure vacuum distillation and freezing, heating and distilling the moisture in the sample in an ultra-low pressure environment, condensing and collecting it in a low-temperature environment to realize the extraction of moisture without fractionation. The instrument has the functions of fault prompt and automatic alarm to avoid wrong sample extraction, ensure the correctness of the sample, and its extraction rate is >98%, and the recovery rate is 99%-101%. However, recent studies have exposed issues related to the effect of vacuum extraction of water on the isotope composition of water (Meißner et al., 2014; Orlowski et al., 2013). But it is still the standard method for extracting soil and plant water in ecological hydrology. And from previous studies, it is still difficult to make post-correction in terms of soil properties or the effects of extraction conditions, because such information is rarely reported, and huge

variability in method details is common (Walker et al., 1994). Before water extraction of soil and plants, the collected samples need to be thawed out of the refrigerator, and each sample bottle should be stuffed with a small cotton ball to prevent water evaporation. When pumping water, we set the pumping time to 150 minutes (180 minutes for plants), the temperature to 190°C, the upper limit of vacuum pressure to 800pa, and the leakage rate to be 0. The water evaporates from the soil, or the plant sample is heated for a specified time and frozen in a liquid nitrogen cold trap. After the extraction is complete, thaw the sample at room temperature and shake it well. Use a 1ml syringe to extract the water sample into a labelled sample bottle, seal it, and wait for the isotope experiment.

[Figure]

**(3)Which uncertainties can be expected due to water extraction?**

Response: The water extraction instrument used is LI-2100 automatic vacuum condensation extraction. The system strictly controls the quality:

(1) Failure prompt and automatic alarm to avoid wrong sample extraction to ensure the sample's correctness.

(2) The traditional classical method is used, and the data is reliable.

(3) Extraction rate: >98%, recovery rate: 99%-101%.

(4) The whole process is automatically completed under the supervision of the control

system.

For plant samples, it is generally appropriate to extract 0.1-0.3 ml of water (for example, 2-3 branches with a length of 3-4 cm), not too much or too little. Too many samples will cause the extraction time to be too long, and it is easy to cause incomplete extraction and affect the results; too few samples are taken, it may be difficult to obtain sufficient water for determination.

Therefore, due to water extraction, we can predict the accuracy of the extracted sample, the reliability of the data obtained by the extraction, and the uncertainty caused by the human operation.

**2. However, this section is very short in the paper although, recent paper show how difficult and partly unreliable these measurement processes still are. I think that with this little information about measurement methods the data set can hardly be used by other research groups or compared with other measurement campaigns.**

Response: Thank you very much for your suggestion. We have added relevant information to the data and methods section of the article, and the added content is as follows:

**4 Data and Methods**

**4.1 Sample collection**

**4.1.1 Collection of precipitation**

We have set up 16 weather stations in the Shiyang River Basin to collect precipitation, including rain barrels used to collect precipitation. The rain barrel is placed in an open outdoor area and consists of rain gear, funnel, water bottle and rain cup. The diameter of the rain gear is 20 cm, and the port of the device is horizontal. The height of the rain opening of the instrument is set to 70 cm from the ground level. A ping-pong ball is placed at the mouth of the funnel, and a layer of paraffin oil is added to the bottom of the containerto prevent isotope fractionation caused by evaporation. Immediately after each precipitation event, transfer the collected liquid precipitation to a 100 ml high-density sample bottle. Solid precipitation must be transferred to a high-density polyethene sample bottle after it becomes liquid water at

room temperature (23°C). The sample bottle is sealed with Parafilm. The polyethene bottle is labelled simultaneously, indicating the date of collecting the sample, the type of precipitation (rain, snow, hail), and the volume of precipitation. Store the collected samples in a refrigerator at about 4°C for later analysis. For multiple precipitation events in one day, various samplings are required.

**4.1.2 Collection of surface water and groundwater**

Polyethene bottles are used to collect surface water (rivers, lakes, reservoirs). Stratified sampling is carried out at different depths (surface layer, middle layer, bottom layer. Groundwater samples were obtained from the groundwater monitoring wells of the Shiyang River Basin Administration, China Hydrological Administration and Gansu Hydrological Administration. The bottle of the sample is sealed with parafilm film and then frozen until the experiment. Simultaneously, the polyethene bottle sample is labelled with the sample's date, sampling depth, and the stream and tributary stream. The collected water samples should be placed in places where the sunlight is not direct to avoid the evaporation of water, which would affect the validity of the data. The samples were taken back to the refrigerator in the laboratory within 10 hours.

**4.1.3 Collection of soil and plant water**

The soil sample is collected at a depth of 100 cm, and samples are taken sequentially at 10cm intervals. The upper reaches of the Shiyang River Basin are mainly clay, and the middle and lower reaches are clay and sand, but sand is the main soil type. Table 2 shows the characteristics of soil in the farmland area of Minqin Oasis. The soil samples collected were divided into two parts, one of which was put into a 50 ml glass bottle. The bottle mouth was sealed with parafilm membrane and transported to the observation station within 10 hours after the sampling date was marked for cryopreservation to test stable isotope data. The other part of the sample was placed in a 50 ml aluminium box and used the drying method to test the soil moisture content since 2019.

Table 2 Basic information of soil samples (Oasis farmland area, Zhu et al., 2021)

| Soil depth (cm) | Clay (%) | Silt (%) | Sand (%) | Soil bulk density (g/cm³) |
|---|---|---|---|---|
| 0-10 | 10.20 | 38.85 | 50.95 | 1.052 |
| 10-20 | 12.94 | 37.76 | 49.30 | 1.194 |
| 20-30 | 10.33 | 44.23 | 45.44 | 1.298 |
| 30-40 | 13.48 | 38.69 | 47.83 | 1.180 |
| 40-50 | 12.01 | 35.09 | 52.90 | 1.140 |
| 50-60 | 11.21 | 42.83 | 45.96 | 1.206 |
| 60-70 | 10.34 | 42.98 | 46.68 | 1.208 |
| 70-80 | 11.09 | 38.96 | 49.95 | 1.106 |
| 80-90 | 11.75 | 37.72 | 50.53 | 1.200 |
| 90-100 | 7.21 | 35.97 | 56.82 | 1.272 |

Plant sample collection: For trees and shrubs, we collect non-green branches, and for herbs, we collect non-green parts at the junction of rhizomes. When sampling, we use scissors to collect vegetation xylem stems, peel off the bark, put them in 50 ml glass bottles, seal them, and freeze them until experimental analysis. Table 3 shows the plant information collected in the Shiyang River Basin

Table 3 Basic information of plant samples

| Sampling points | Vegetation types | Sample size |
|---|---|---|
| BDZ | Agropyron cristatum | 30g ±0.5 |
| CQQ | Corn (leaf), reed, jujube (Branches), dryland willow (Branches) | 30g-100g |
| DT | Reed | 30g±0.5 |
| DTX | Spring wheat (stem), corn (root, stem, leaf) | 30g±0.5 |
| HJX | Willow (Branches) | 100g±0.5 |
| HLD | Qinghai Spruce (Branches) | 100g±0.5 |
| HLZ | Qinghai Spruce (Branches) | 100g±0.5 |
| WWPD | Corn (leaf), wheat (leaf) | 30g±0.5 |
| XYWG | Poplar (Branches), wheat | 100g±0.5 |
| YXB | Corn (leaf) | 30g±0.5 |
| SWX | Corn, wheat (leaf) | 30g±0.5 |
| LLL | Salsola purpurea | 30g±0.5 |

**4.1.4 Collection of meteorological data**

The local meteorological data were obtained and recorded during the sampling period by the automatic weather stations (watchdog 2000 series weather stations) erected near the sample plot. Meteorological data include temperature(℃), relative humidity(%), atmospheric pressure(hPa), dew point temperature(℃) and precipitation amount(mm).

**4.2 Experiment analysis**

**4.2.1 Water extraction experiment**

We use vacuum condensation to extract soil and plant water. The extraction equipment used is LI-2100 automatic vacuum condensation extraction equipment. Before water extraction is performed on the soil and plants, the collected samples need to be taken out of the refrigerator to thaw, and each sample bottle should be stuffed with a small ball of cotton to prevent the water from evaporating. When extracting water, we set the extraction time to 150 minutes (180 minutes for plants), the temperature to 190°C, the upper limit of the vacuum pressure to 800pa, and the leakage rate to 0. The water evaporates from the soil or plant sample by heating it for a specified time and then freezes it in a liquid nitrogen cold trap. After the extraction is completed, the sample is thawed at room temperature and shaken. Use a 1ml syringe to extract the water sample into a labelled sample bottle, seal it and wait for the isotope experiment.

**4.2.2 Isotope experiment**

All the collected water samples were analyzed in the stable isotope laboratory of Northwest Normal University using liquid water isotope analysis (DLT-100, Los Gatos Research, USA). Each water sample and isotope standard sample were injected six times in a row. We discarded the first two injections and used the average of the last four times as the final result to eliminate the instrument memory effect. The result of the isotope measurement is expressed by the symbol "δ" and expressed in thousandths of the difference relative to the Vienna Standard Mean Ocean Water (Craig, 1961):

$$\delta_{sample}\ (‰) = \left[ \left( \frac{R_{sample}}{R_{v\text{-}smow}} \right) - 1 \right] \times 1000$$

(1)

In the formula, Rsample is the ratio of $^{18}O/^{16}O$ or $^2H/^1H$ in the collected sample, Rv-smow is the ratio of $^{18}O/^{16}O$ or $^2H/^1H$ in the Vienna standard sample.The analytical accuracy of $\delta^2H$ and $\delta^{18}O$ are ±0.6‰ and ±0.2‰, respectively.

4.2.3 Modification of plant water isotope data

Suppose the water sample contains compounds with the same absorption

characteristics of the same wavelength. In that case, it will lead to errors in the measurement of the laser liquid water analyzer, and the most likely pollutants to cause errors are methanol and ethanol. So using deionized water with different concentrations of pure methanol and ethanol, the combination of Los Gatos company LWIA - Spectral Contamination Identifier v1.0 Spectral analysis software (NB) to determine methanol and ethanol (BB) pollution degree of spectrum measurement, establishing the δD and δ¹⁸O correction method for the spectra of pollution (Meng et al., 2012; Liu et al., 2013). In the correction process, methanol and ethanol solution concentration configuration were similar to Meng's experiment (2012). Correction results for methanol its broadband measurements of NB metric logarithmic respectively with ΔδD and Δδ¹⁸O are significantly quadratic curve relationship, respectively is:

$$\Delta \delta D = 0.018(\ln NB)^3 + 0.092(\ln NB)^2 + 0.388\ln NB + 0.785(R^2 = 0.991,\ p > 0.0001) \qquad (2\text{-}1)$$

$$\Delta \delta^{18}O = 0.017(\ln NB)^3 - 0.017(\ln NB)^2 + 0.545\ln NB + 1.356(R^2 = 0.998,\ p < 0.0001) \qquad (2\text{-}2)$$

Its broadband measurements for ethanol correction results in BB metric and ΔδD and Δδ¹⁸O, a quadratic curve and linear relationship respectively, are:

$$\Delta \delta D = -85.67BB + 93.664(R^2 = 0.747, p = 0.026)(BB < 1.2) \qquad (2\text{-}3)$$

$$\Delta \delta^{18}O = -21.421BB^2 + 39.935BB - 19.089(R^2 = 0.769,\ p < 0.012) \qquad (2\text{-}4)$$

**4.3 Data quality**

It has always been a difficult problem to control the experimental error to the minimum. We use Manner-Kendall to test meteorological and hydrological data, eliminate abnormal values, and use interpolation to obtain vacant values. For the isotope data, we first use the LIMA software to check the original isotope data. We stipulate that when one or more of the 6 data of a sample is marked in red, we call the sample an error sample, even though the sample is in the display on the liquid water

analyzer is normal, and we will re-experiment the sample until it passes the LIMA software verification. Then we will use SPSS software to check the normality of the obtained isotope data. At present, our experimental errors mainly come from the following aspects:

4.3.1 Sample collection

The error caused by precipitation sample collection mainly comes from the sampling personnel's failure to transfer the rainwater in the rain gauge or mixing two or more rainwaters after each precipitation event, which will cause the rainwater in the rain gauge to evaporate and make The measured value deviates greatly from the actual value.

The error caused by the collection of vegetation samples mainly comes from the slow speed of collecting samples, and the vegetation is exposed to the air for a long time, which causes the vegetation to fractionate water.

The error in collecting soil samples is that we do not know much about the influence of soil microbial activities on the extracted water isotope results. Therefore, we did not pay too much attention to the microorganisms in the soil when collecting the samples. As a result, the collected soil samples contained many microorganisms, which greatly impacted the data results.

4.3. 2 Experiment

The experimental error is mainly because we set the same moisture extraction parameters for samples with different soil characteristics. It is difficult to make post-mortem corrections for soil properties or the effects of extraction conditions because such information is rarely reported, and massive variability in method details is common (Walker et al., 1994). In addition, there are still measurement uncertainties during the extraction of water, which also come from the loss of water vapor during the vacuum of the extraction system and the non-temperature heating temperature, which will lead to experimental errors.

Our calibration of plant sample data only considers methanol and ethanol pollution, but the plant and soil water extracts may contain various other pollutants, leading to experimental errors. In addition, studies have shown that the mismatch

between xylem and plant water sources is due to the fractionation of isotopes in the process of water absorption (Poca et al., 2019), which questioned the fact that plants do not undergo fractionation during the process of water absorption (Porporato, 2001; Meissne et al., 2014) this traditional view. However, in the experiment, we still use traditional viewpoints as assumptions, leading to experimental errors.

**3. Soil properties can highly influence stable water isotope measurements (see for instance the work by Orlowski). However, information about the soil properties are not provided.**

Response: Thank you very much for your suggestion. We have added the content of soil properties in the section 4.1.3. and the added content is as follows:

The upper reaches of the Shiyang River Basin are mainly clay, and the middle and lower reaches are clay and sand, but sand is the main soil. Table 2 shows the characteristics of soil in the farmland area of Minqin Oasis.

Table 2 Basic information of soil samples(Oasis farmland area, Zhu et al., 2021)

| Soil depth (cm) | Clay (%) | Silt (%) | Sand (%) | Soil bulk density (g/cm$^3$) |
|---|---|---|---|---|
| 0-10 | 10.20 | 38.85 | 50.95 | 1.052 |
| 10-20 | 12.94 | 37.76 | 49.30 | 1.194 |
| 20-30 | 10.33 | 44.23 | 45.44 | 1.298 |
| 30-40 | 13.48 | 38.69 | 47.83 | 1.180 |
| 40-50 | 12.01 | 35.09 | 52.90 | 1.140 |
| 50-60 | 11.21 | 42.83 | 45.96 | 1.206 |
| 60-70 | 10.34 | 42.98 | 46.68 | 1.208 |
| 70-80 | 11.09 | 38.96 | 49.95 | 1.106 |
| 80-90 | 11.75 | 37.72 | 50.53 | 1.200 |
| 90-100 | 7.21 | 35.97 | 56.82 | 1.272 |

**4. The study talks about "xylem" measurement. However, it remains unclear whether stem, branches or twigs were used. In addition, the tree / plant species remain unknown.**

Response: Thank you very much for your suggestion. We added the plant sample information table in section 4.1.3, the added table is as follows:

Table 3 Basic information of plant samples

| Sampling points | Vegetation types | Sample size |
|---|---|---|
| BDZ | Agropyron cristatum | 30g ±0.5 |

| | | |
|---|---|---|
| CQQ | Corn (leaf), reed, jujube (Branches), dry willow (Branches) | 30g-100g |
| DT | Reed | 30g±0.5 |
| DTX | Spring wheat (stem), corn (root, stem, leaf) | 30g±0.5 |
| HJX | Willow (Branches) | 100g±0.5 |
| HLD | Qinghai Spruce (Branches) | 100g±0.5 |
| HLZ | Qinghai Spruce (Branches) | 100g±0.5 |
| WWPD | Corn (leaf), wheat (leaf) | 30g±0.5 |
| XYWG | Poplar (Branches) | 100g±0.5 |
| YXB | Corn (leaf) | 30g±0.5 |
| SWX | Corn, wheat (leaf) | 30g±0.5 |
| LLL | Salsola purpurea | 30g±0.5 |

**4. I miss the information about the delta notation of isotopes. I would assume it is in reference to the Vienna Standard.**

Response: Thank you very much for your suggestion. We have added isotope laboratory analysis to the data and methods, and added the following content:

All the collected water samples were analyzed in the stable isotope laboratory of Northwest Normal University using liquid water isotope analysis (DLT-100, Los Gatos Research, USA). Each water sample and isotope standard sample were injected six times in a row. We discarded the first two injections and used the average of the last four times as the final result to eliminate the instrument memory effect. The result of the isotope measurement is expressed by the symbol "$\delta$" and expressed in thousandths of the difference relative to the Vienna Standard Mean Ocean Water (Craig, 1961):

$$\delta_{sample}\ (‰) = \left[\left(\frac{R_{sample}}{R_{v\text{-}smow}}\right) - 1\right] \times 1000$$

(1)

In the formula, Rsample is the ratio of $^{18}O/^{16}O$ or $^2H/^1H$ in the collected sample, Rv-smow is the ratio of $^{18}O/^{16}O$ or $^2H/^1H$ in the Vienna standard sample.The analytical accuracy of $\delta^2H$ and $\delta^{18}O$ are ±0.6‰ and ±0.2‰, respectively.

**Specific comments**

**1. Abstract: I miss the very important information about the timescale.**

Response: We have added timescale information to the Abstract. The revised sentence is as follows:

We have established a stable water isotope monitoring network from 2015 to 2019 in the Shiyang River Basin in China'arid northwest.

**2. L15: I think it would improve the readability when first the water bodies and then the areas would be explained. However, you also could mention that you measured stable water isotopes earlier in the abstract.**

Response: We first introduce the water body and then the area according to your suggestion. The revised sentence is as follows:

All data in the data set are differentiated by water body types (precipitation, river water, lake water, groundwater, soil water, plant water). The monitoring station covers the upper, middle and lower reaches of the river basin, with six observation systems: river source area, oasis area, reservoir canal system area, oasis farmland area, ecological restoration area, and salinized area.

**3. L20: Maybe use the term stable WATER isotopes here. Make sure to use one term consistently in your text. Later you used the term stable water isotopes, too.**

Response: We have revised it according to your suggestion, and the revised sentence is as follows:

The data set includes stable water isotope data, meteorological data and hydrological data in the Shiyang River Basin.

**5. L20: please define roughly which meteorological and hydrological data is connected to your data set. Additionally, no meteorological or hydrological data are found in the data files.**

Response: We added data related to meteorological and hydrological data sets in Section 5.1 based on your suggestions. The revised sentence is as follows:

Meteorological data includes temperature (℃), relative humidity (%), atmospheric pressure (hPa), dew point temperature (℃) and precipitation (mm). Hydrological data includes annual runoff (m³/s), daily average flow (m³/s), daily average water level (m), and water inflow and changes in Qingtu Lake.

We have added meteorological data and hydrological data to the data file. The new data set are openly available in Zhu, Guofeng (2021), "Stable water isotope monitoring network of different water bo

dies in Shiyang River Basin, a typical arid river in China (Supplemental Edition 2021 0808)", Mendeley Data, V1, doi: 10.17632/d5kzm92nn3.1.

**6. L34: use D or 2H consistent in your paper, please.**

Response: Thank you very much for your suggestion. We have unified the hydrogen in the article as D.

**7. L37: please generally check your citation style! Sometimes the ";" symbols are weird looking sometimes there are too many symbols e. g. L47.**

Response: We have modified the citation marks in the text and deleted the redundant citation marks in the text.

**8. L37ff: I would be nice to see some more specific examples which can be related to the data presented here. Moreover, I am missing completely a critical view on the usage of stable water isotopes and their uncertainties here. E. g. unintended fractioning processes as well as troubles due to measurement methods!**

Response: We have added relevant content based on your suggestions, and the added content is as follows:

For example, Meißner et al. (2014) emphasized that the change of $\delta^{18}O$ largely depends on the soil type (Araguás-Araguás et al., 1995). Orlowskii et al. (2016) showed that incomplete water extraction in the cryogenic distillation process might fractionate water isotopes. Therefore, clay requires a longer extraction time and temperature to reduce the fractionation effect in the extraction process. In addition, a study by Sofer and Gat in 1975 showed that the formation of hydrated spheres around cations in aqueous solutions would fractionate the oxygen isotopes of the water. Gaj et al. (2017) showed that the isotope characteristics are biased due to a process different from Rayleigh distillation that we cannot reduce the effect caused by the mineral-water interaction entirely.

**9. L47: Christophe is a first name! Also, this article is about oceans I am not sure how well this fits your paper.**

Response: Thank you very much for your suggestion. We have added other references, the added references are as follows:

Liu, J., Gao, Z., Wang, M., Li, Y., Yu, C., Shi, M., Zhang, H., and Ma, Y.: Stable isotope characteristics of different water bodies in the Lhasa River Basin. Environmental Earth Sciences, 78(3), 71. https://doi.org/10.1007/s12665-019-8078-6, 2019.

Yang, Y., Xiao, H., Qin, Z., and Zou, S.: Hydrogen and oxygen isotopic records in monthly scales variations of hydrological characteristics in the different landscape zones of alpine cold regions. Journal of Hydrology, 499, 124-131. https://doi.org/10.1016/j.jhydrol.2013.06.025, 2013.

**10. L47: I could find the article of Zhang et al., 2015 only in non-English language. Please provide another link in the bibliography or provide further information.**

Response: We have revised its references:

Liu, J., Gao, Z., Wang, M., Li, Y., Yu, C., Shi, M., Zhang, H., and Ma, Y.: Stable isotope characteristics of different water bodies in the Lhasa River Basin. Environmental Earth Sciences, 78(3), 71. https://doi.org/10.1007/s12665-019-8078-6, 2019.

Yang, Y., Xiao, H., Qin, Z., and Zou, S.: Hydrogen and oxygen isotopic records in monthly scales variations of hydrological characteristics in the different landscape zones of alpine cold regions. Journal of Hydrology, 499, 124-131. https://doi.org/10.1016/j.jhydrol.2013.06.025, 2013.

**11. L51-L52: I think other more important paper would suit this sentence much better! Also, Metthew is a first name!**

Response: Thank you very much for your suggestion. We have added other references, the added references are as follows:

Risi, C., Landais, A., Winkler, R., and Vimeux, F.: Can we determine what controls the spatio-temporal distribution of d-excess and $O^{17}$-excess in precipitation using the LMDZ general circulation model? Climate of the Past, 9(5): 2173-2193, https://doi.org/10.5194/cp-9-2173-2013, 2013.

Pfahl, S., and Wernli, H.: Air parcel trajectory analysis of stable isotopes in water vapor in the eastern mediterranean. Journal of Geophysical Research Atmospheres, 113(D20), https://doi.org/ 10.1029/2008JD009839, 2008.

**12. L62: I think that this should be much more critical! There are many drawback and uncertainties using stable water isotopes which should be at least briefly mentioned. See: Poca et. al 2019 (doi: 10.1007/s11104-019-04139-1); von Freyberg (doi: 10.1111/2041-210X.13461) and many more!**

Response: Thank you very much for your suggestion. We have added the shortcomings and uncertainties of using stable water isotopes, the added content is as follows:

Studies have shown that physicochemical soil properties may cause the fractionation of hydrogen and oxygen in soil water (Meißner et al., 2014). Because we do not know whether the unstable hydrogen fraction during the low-temperature extraction process will cause isotope fractionation (Orlowski et al., 2016). In addition, we know little about the effect of soil microbial activity on the extracted water isotope results (Orlowski et al., 2018). However, from previous studies, it is still difficult to make post-correction in terms of soil properties or the effects of extraction conditions because such information is rarely reported, and huge variability in method details is common (Walker et al., 1994). We have always known that these "problems" exist, but water vacuum extraction is still the standard method for extracting soil and plant water in ecological hydrology (Ingraham et al., 1992). In most plants, the isotopic composition of water does not change due to root absorption and transport through the stem xylem (White et al., 1985). However, more and more studies have shown a difference between the isotope composition of xylem water and plant water sources (Poca et al., 2019) and the fractionation can occur along the root water absorption pathway. This fact does not make the isotope method in the soil-plant-atmosphere useless to track water in the continuum, and $\delta D$ and $\delta^{18}O$ can still be used similarly to study water absorption of various plants (Poca et al., 2019).

**13. L80: change ues to use, please.**

Response: Thank you very much for your suggestion. We have changed ues to use.

**14. L76 ff: This is very similar to the abstract. I miss your motivation for the study and why the data of this specific region is important. In the abstract you mention something about ecological importance etc.**

Response: Thank you very much for your suggestion. We added research motivation and explained the importance of Shiyang River Basin data. The added content is as follows:

The Shiyang River Basin is one of the earliest developed river basins in Northwest China and is considered one of the most severely over-exploited inland river basins in China (Kang et al., 2004). It is one of the most prominent areas of water conflict and ecological environment problems. This paper compiles the stable water isotope data of the Shiyang River Basin from 2015 to 2019 and its corresponding meteorological and hydrological data into a data set. This work will help clarify the local water cycle in the Shiyang River Basin and the impact of human activities on agricultural production, analysis the development trend of the Shiyang River Basin under the background of global warming, provide a certain scientific reference for other semi-arid regions in the world, and promote ecological restoration in the arid regions.

**15. L93: what do the coordinate show / mean?**

Response: This geographic coordinate represents the location of the Shiyang River Basin. It should have been placed in the first paragraph of the study area. We have modified it. The revised sentence is as follows:

The Shiyang River Basin (101°41'-104°16'E and 36°29'-39°27'N) is located in the eastern section of Qilian Mountain and Hexi Corridor, and it is the third-largest river in the flowing water system in the Hexi Corridor of Gansu Province. The topography of the Shiyang River Basin slopes sharply from southwest to northeast, with Qilian Mountains in the south, alluvial plains and Gobi in the middle, and flood plains and deserts in the north (Zhu et al., 2020). The river is about 250 km long, and the basin area is $4.16 \times 10^4 \, km^2$.

**16. L94: numbers such as average discharge in m3/s would be more helpful, I think. If seasonality is important, show average values from summer / winter or**

**dry / rain season.**

Response: We have added information related to the flow of the Shiyang River, and the added content is as follows:

The annual average flow rate of the Shiyang River is 8.33 m³/s, of which the multi-year average flow rate in spring is 7.55 m³/s, the annual average flow rate in summer is 15.24 m³/s, the annual average flow rate in autumn is 8.66 m³/s, and the average flow rate in winter is 1.89 m³/s.

**17. L96 what does average annual temperature below 6 C mean?**

Response: The annual average temperature below 6°C means that the Qilian Mountains in the south of the Shiyang River have perennial glacier tundra, which is the runoff formation area of the Shiyang River Basin and is of great significance to the Shiyang River Basin in the arid area. Therefore, it is of great significance to protect the glaciers and forest and grassland belts in the runoff-forming area of the upper Qilian Mountains. The protection of glaciers and snow lines in the Qilian Mountains is more consistent with climate change. Human activities to reduce the emission of black carbon substances as much as possible can slow down the retreat of regional glaciers and snow lines, but it takes a long process.

**18. L126: rain collection is slightly unclear to me. How did you store the water to prevent evaporation before measurement? This would be crucial to know for further discussion of data quality and possible uncertainties.**

Response: We have set up 16 weather stations in the Shiyang River Basin to collect precipitation, including rain barrels used to collect precipitation (see the picture below). The rain barrel is placed in an open outdoor area and consists of rain gear, funnel, water bottle and rain cup. The diameter of the rain gear is 20 cm, and the port of the device is horizontal. The height of the rain opening of the instrument is set to 70 cm from the ground level. A ping-pong ball is placed at the mouth of the funnel, and a layer of paraffin oil is added to the bottom of the containerto prevent isotope fractionation caused by evaporation. Immediately after each precipitation event, transfer the collected liquid precipitation to a 100 ml high-density sample bottle. Solid precipitation must be transferred to a high-density polyethene sample bottle after it

becomes liquid water at room temperature (23°C). The sample bottle is sealed with Parafilm. The polyethene bottle is labelled simultaneously, indicating the date of collecting the sample, the type of precipitation (rain, snow, hail), and the volume of precipitation. Store the collected samples in a refrigerator at about 4°C for later analysis. For multiple precipitation events in one day, various samplings are required.

[Figure]

**19. L149 bold!**

Response: We have made changes, and the revised sentence is as follows:

**4.1.3 Collection of soil and plant water**

**20. L150 unclear where soils samples are taken. What does "sequentially at 10 cm intervals" mean?**

Response: Our sampling point information is shown in Table 1. The soil sampling points are SCG, HJX, HLZ, WWPD, DT, HYSSK, CQQ, MQDQ, QTH, SWX, DTX, YXB, SYQ, JCLK, HLD, BGH, LKS, and XYWG. We sample the soil at 10 cm intervals. The total depth of the collected soil is 100 m. A total of 10 layers of soil samples were collected, which were 0-10 cm, 10-20 cm, 20-30 cm, 30-40 cm, 40-50 cm, 50-60 cm, and 60-70 cm, 70-80 cm, 80-90 cm, 90-100 cm.

**21. L155 a new paragraph would fit here.**

Response: We have made changes according to your suggestion, and the revised sentence is as follows:

The soil sample is collected at a depth of 100cm, and samples are taken sequentially at 10cm intervals. The upper reaches of the Shiyang River Basin are mainly clay, and the middle and lower reaches are clay and sand, but sand is the main soil type. Table 2 shows the characteristics of soil in the farmland area of Minqin Oasis. The soil samples collected were divided into two parts, one of which was put

into a 50 ml glass bottle. The bottle mouth was sealed with parafilm membrane and transported to the observation station within 10 hours after the sampling date was marked for cryopreservation to test stable isotope data. The other part of the sample was placed in a 50 ml aluminium box and used the drying method to test the soil moisture content since 2019.

Plant sample collection: For trees and shrubs, we collect non-green branches, and for herbs, we collect non-green parts at the junction of rhizomes. When sampling, we use scissors to collect vegetation xylem stems, peel off the bark, put them in 50ml glass bottles, seal them, and freeze them until experimental analysis. Table 3 shows the plant information collected in the Shiyang River Basin

**22. L155 did you use branches / twigs or stem xylem? How much (better in gram instead of a volume). Which tree species (conifers / hardwood)? Here, a lot of important information is missing!**

Response: We have added a plant sample information table:

Table 3 Basic information of plant samples

| Sampling points | Vegetation types | Sample size |
|---|---|---|
| BDZ | Agropyron cristatum | 30g ±0.5 |
| CQQ | Corn (leaf), reed, jujube (Branches), dry willow (Branches) | 30g-100g |
| DT | Reed | 30g±0.5 |
| DTX | Spring wheat (stem), corn (root, stem, leaf) | 30g±0.5 |
| HJX | Willow (Branches) | 100g±0.5 |
| HLD | Qinghai Spruce (Branches) | 100g±0.5 |
| HLZ | Qinghai Spruce (Branches) | 100g±0.5 |
| WWPD | Corn (leaf), wheat (leaf) | 30g±0.5 |
| XYWG | Poplar (Branches) | 100g±0.5 |
| YXB | Corn (leaf) | 30g±0.5 |
| SWX | Corn, wheat (leaf) | 30g±0.5 |
| LLL | Salsola purpurea | 30g±0.5 |

**23. L164: I think the abbreviations are everything else but "easy to understand and use". I would prefer abbreviation which include somehow sampling type (i. e. river, precipitation, soil) and the site.**

Response: We have revised this sentence, and the revised sentence is as follows:

Table 1 lists the complete names, abbreviations and corresponding

meteorological parameters of each sampling point, which is convenient for readers to correspond with the data set.

We have modified Table 1 according to your suggestion, and the revised Table 1 is as follows:

Table 1 List of basic parameters

| Abbreviation | Full name | Longitude | Latitude | Elevation (m) | Average annual precipitation ( mm) | Sampling type (abbreviation) | Sampling type (full name) | Location |
|---|---|---|---|---|---|---|---|---|
| QHLYXM | Qinghai Forestry Project | 101°51' | 37°32' | 3899 | - | hs | river water | a |
| MK | Colliery | 101°51' | 37°33' | 3647 | 595.10 | hs | precipitation | a |
| LXWL | Winding Road | 101°50' | 37°34' | 3305 | - | hs | river water | a |
| SDHHC | Tunnel Junction | 101°50' | 37°34' | 3448 | - | hs | river water | a |
| BDZ | Transformer Substation | 101°51' | 37°33' | 3637 | - | tr, zw, hs | soil, plant, river water | a |
| NQ | Ningqian | 101°49 | 37°37' | 3235 | - | hs | river water | a |
| SCG | Ningtanhe Middle East branch mixed water | 101°50' | 37°38' | 3068 | - | hs, js, tr | river water, precipitation, soil | a |
| MTQ | Wood Bridge | 101°53' | 37°41' | 2741 | - | hs | river water | a |
| SCLK | Three-way Intersection | 101°55' | 37°43' | 2590 | - | hs | river water | a |
| JTL | Nine Ridge | 102°02' | 37°51' | 2267 | - | dxs | groundwater | a |
| WGQ | The Bridge of the Cultural Revolution | 102°07' | 37°53' | 2174 | - | hs | river water | a |
| XYSK | Xiying Reservoir | 102°12' | 37°54' | 2058 | - | hs | river water | c |
| XYZ | Xiying Town | 102°26' | 37°58' | 1748 | 491.35 | js | precipitation | c |
| GGKFQ | Reform and Opening Bridge | 101°58' | 37°46' | 2590 | - | hs | river water | c |
| HJX | Huajian Township | 102°00' | 37°50' | 2390 | 262.64 | hs, dxs, js, tr | river water, groundwater, precipitation, soil | c |
| WW | Wuwei | 102°37' | 37°53' | 1581 | 300.14 | hs | river water | c |
| HLZ | Ranger Stations | 101°53' | 37°41' | 2721 | 469.44 | hs, js, tr, zw, dxs | river water, precipitation, soil, plant, | a |

| | | | | | | | | |
|---|---|---|---|---|---|---|---|---|
| LLL | Lenglong Ridge | 101°28' | 37°41' | 3500 | 350.34 | js | precipitation | a |
| ZZXL | Zhuaxi Xiulong | 103°20' | 37°18' | 3556 | 500.17 | js | precipitation | d |
| JDT | Jiudun Beach | 102°45' | 38°07' | 1464 | - | js | precipitation | d |
| SCG | Shangchigou | 102°25' | 38°03' | 2400 | 377.13 | js, hs, dxs | precipitation, river water, groundwater | d |
| WWPD | Wuwei Basin | 102°42' | 38°06' | 1467 | - | js, dxs, tr, zw | precipitation, groundwater, soil, plant | d |
| DT | Dongtan | 102°47' | 38°16' | 1434 | 240.05 | hs,tr, zw | river water, soil, plant | e |
| HYSSK | Hongyashan Reservior | 102°53' | 38°24' | 1416 | 100.17 | hs, dxs, tr | river water, groundwater, soil | f |
| CQQ | Caiqi Bridge | 102°45' | 38°13' | 1443 | 300.26 | dxs, hs, tr, zw | groundwater, river water, soil, plant | d |
| XJG | Xiajiangou | 102°42' | 38°07' | 1200 | 110.18 | dxs | groundwater | d |
| HGG | Hongqi Valley | 102°50' | 38°21' | 1421 | 113.16 | js, dxs | precipitation, groundwater | d |
| BHZ | Protection Station | 102°29' | 38°09' | 2787 | - | dxs | groundwater | d |
| BDC | Beidong Township | 103°02' | 38°32' | 1367 | 155.45 | dxs | groundwater | g |
| XXWGZ | Xiyin Wugou Township | 102°58' | 38°29' | 1393 | - | dxs | groundwater | d |
| MQBQ | Minqin Dam | 103°08' | 39°02' | 1400 | 113.19 | tr | soil | d |
| QTH | Qingtu Lake | 103°36' | 39°03' | 1313 | 110.79 | js, dxs, ls, tr | precipitation, groundwater, lake water, soil | h |
| SWX | Suwu Township | 103°05' | 38°36' | 1372 | 155.84 | dxs, tr, zw, hs | groundwater, soil, plant, river water | d |
| DTX | Datan Township | 103°14' | 38°46' | 1349 | - | js, dxs, soi, zw, hs | precipitation, groundwater, soi, plant, river water | g |
| YXB | Yangxia Dam | 102°41' | 38°01' | 1489 | - | js, dxs, tr, zw | precipitation, groundwater, soil, plant | d |
| XBZ | Xuebai Toen | 103°01 | 38°32' | 1387 | - | js | precipitation | b |
| SYQ | Laboratory Area | 102°22' | 37°42' | 2438 | - | hs, tr | river water, soil | b |

| | | | | | | | |
|------|------------------------|---------|--------|-------|--------|-------------|------------------------------------|---|
| XCL | Small Valley | 102°24' | 37°43' | 2267 | - | hs | river water | b |
| JCLK | Intersection | 102°20' | 37°41' | 2544 | - | hs, tr | river water, soil | b |
| QSHSY | Spring River | 102°22' | 37°38' | 2747 | - | qs | spring water | b |
| HLD | Confluence | 102°26' | 37°44' | 2146 | - | hs, tr, zw | river water, soil, plant | b |
| QXZ | Meteorological Station | 102°20' | 37°42 | 2543 | 510.56 | js, dxs | precipitation, groundwater | b |
| BGH | Binggou River | 102°17' | 37°40' | 2872 | - | hs, tr, | river water, soil | b |
| NCHHLH | South Nancha River | 102°26' | 37°43' | 2163 | - | hs | river water | b |
| LKS | Two Pine | 102°17' | 37°40' | 2832 | - | hs, tr | river water, soil | b |
| NYSKRK | Nanying Reservoir | 102°29' | 37°47' | 1955 | 330.16 | hs | river water | b |
| SGZZ | Sigou stckade | 102°23' | 37°40' | 2492 | 675.54 | hs | river water | b |
| JZGD | Construction Site | 102°25' | 37°41' | 2303 | - | hs | river water | b |
| QLX | Qilian Township | 102°42' | 38°08' | 3394 | 300.15 | js, qs | precipitation, spring water | d |
| XYWG | Xiying Wugou | 102°10' | 37°53' | 2097 | 197.67 | hs, js, tr, zw | river water, precipitation, soil, plant | c |
| HSH | Hongshui River | 102°45' | 38°13' | 1454' | - | hs | river water | d |
| XCL | Small village | 102°24' | 37°43' | 2267 | - | hs | river water | b |
| YHRJ | A family | 102°20' | 37°42' | 2543 | - | hs | river water | b |

**24. L168: the corresponding "weather station file" was not found in the data set. Also, should have multiple sampling types not have the same abbreviation in case someone will merge the different data sets.**

Response: We have added the corresponding meteorological data set, which can be found in Zhu, Guofeng (2021), " Stable water isotope monitoring network of different water bodies in Shiyang River Basin, a typical river in China (Supplemental Editional 20210808)", Mendeley Data, V1, doi: 10.17632/d5kzm92nn3.1 and we have modified the abbreviation of the sampling type in the data set.

**25. L197-L198: please, define what did you do with the software for data quality check.**

Response: We use the LIMA software to check the original isotope data. We stipulate

that when one or more of the 6 data of a sample is marked in red, we call the sample an error sample, even though the sample is in the display on the liquid water analyzer is normal, and we will re-experiment the sample until it passes the LIMA software verification

**26.  L205 very unclear.**

Response: We have rewritten this sentence, and the revised sentence is as follows:

The vegetation samples should be collected quickly, otherwise the vegetation will be exposed to the air, which will cause the water in the vegetation to be fractionated and affect the quality of the data. After sampling, the sample bottle needs to be sealed with parafilm immediately, and then frozen and stored.

**27.  L213: you mention that you should "select the wrong samples". How do you define "wrong samples"?**

Response: We use the LIMA software to check the original isotope data. We stipulate that when one or more of the 6 data of a sample is marked in red, we call the sample an error sample, even though the sample is in the display on the liquid water analyzer is normal.

**28.  L247: I think that information about the water level are not of interested here. Discharge data would be more interesting to give the reader an idea about the river properties.**

Response: We have modified this part of the content, and the revised content is as follows:

According to the five hydrological observation stations in the Shiyang River Basin, the average annual flow of the Shiyang River Basin from 2015 to 2019 was 9.72 $m^3$/s. Among them, the annual flow in 2015, 2016, 2017, 2018, and 2019 were 9.56 $m^3$/s, 10.67 $m^3$/s, 10.11 $m^3$/s, 7.18 $m^3$/s, 11.03 $m^3$/s, and the flow in summer and autumn was greater than that in winter and spring. This is related to the precipitation of the Shiyang River concentrated in summer. Taking 2015 as an example, the maximum flow of the Shiyang River Basin was 57.0 $m^3$/s, which appeared on July 5. The minimum flow rate was 0.55 $m^3$/s, which appeared on June 11. The annual

average flow is 9.56 m³/s, the annual runoff is 3.016×10⁸ m³/s, the runoff modulus is 0.936 × 10⁻³ m³/(S.km²), and the runoff depth is 29.5 mm. The largest flood volume1day, 3days, 7days, 15days, 30days, and 60days occurred on July 5, July 4, July 4, July 2, July 2, and June 22.

[Figure]

Figure 4 The daily flow of Shiyang River in 2015-2019 (black shaded winter, black diagonal line represents spring, gray shaded summer, gray grid represents autumn)

**29. L286 I guess this conclusion is not fitting well since you would not expect soil water to be recharged by plant water.**

Response: What we wrote in the manuscript is "The average value of $\delta^{18}O$ and $\delta D$ of soil water is between plant water and precipitation, but closer to precipitation (Table 2), indicating that the soil is mainly recharged by precipitation." What we express means is soil water is mainly replenished by precipitation, and we did not mention in the manuscript that soil water is recharged by plant water.

**30. L293 What does 4.4 reference to?**

Response: 4.4 refers to the absolute value of the lake water $\delta^{18}O$ variation coefficient.

**31. L294 what do these numbers mean? Unit?**

Response: The numbers represent the absolute value of the $\delta^{18}O$ variation coefficient of groundwater, river water, soil water, precipitation and plant water, namely: the absolute value of the $\delta^{18}O$ variation coefficient of groundwater is 0.08, the absolute value of the $\delta^{18}O$ variation coefficient of river water is 0.11, the absolute value of the

$\delta^{18}O$ variation coefficient of soil water is 0.37, the absolute value of the $\delta^{18}O$ variation coefficient of precipitation is 0.71, and the absolute value of the $\delta^{18}O$ variation coefficient of plant water is 2.54.

**32. L317 the outlook is missing or it is at least very short.**

Response: We have rewritten outlook according to your suggestion, and the revised content is as follows:

This study collected the stable isotope values of different water bodies in the Shiyang River Basin of China's arid inland river basin from 2015 to 2019. As of 2019, we have collected 6,760 isotope data one by one and formed a data set . Through the analysis of different water stable isotopes in the Shiyang River Basin, the main conclusions reached are: (1) The slope and intercept of the LMWL in the Shiyang River Basin are both smaller than GMWL, which is the same as that of the study area located in the arid inland of Northwest China. When raindrops fall to the surface, they are subject to strong evaporation. (2) The stable isotopes in the Shiyang River Basin show periodic fluctuations, depleted in winter and spring and enriched in summer and autumn. (3) The fluctuation of the stable isotope of lake water is greater than that of other water bodies, and the coefficient of variation is the largest. This is related to the large seasonal differences in the evaporation of Qingtu Lake. (4) The change of groundwater lags behind that of surface water. (5) The main source of replenishment in the Shiyang River Basin is precipitation. This data set provides a new data basis for studying the stable water isotopes of different water bodies in the inland river basins of China. Through these data, we can study the water conservancy connection between different water bodies, analyze the regional water cycle, and study the impact of human activities on farmland, to provide certain guidelines for the rational use of water resources in arid regions.

The accuracy of the isotope data we obtained is not high because we ignored many factors that would cause data errors during the experiment. If we did not consider the influence of the activity of microorganisms in the soil on the isotope data when collecting soil samples, we set the parameters of samples with different soil characteristics to be the same, and we only considered the errors of methanol and

ethanol when calibrating the data. However, the ecological hydrology and soil science communities still lack standard norms and suggestions on these issues. Studies have shown that the new continuous in situ measurement of the isotopic composition of soil and plant water may overcome the problem of isotopic fractionation observed when we extract water. However, the fractionation problems caused by different soil characteristics during the experiment will only be studied through a large number of experiments in the future.

As our field observations proceed, our data set will be updated year by year. To improve the dataset, we encourage users of the dataset to contact the author for suggestions.

**33. Figure. 1: It is unclear whether north is on top of the maps; Big map: It is hard to find point b and point a is hardly visible. The scale of elevation shows very random numbers.**

Response: We have added a north arrow to the picture to improve the clarity of the picture and facilitate the identification of the specific positions of the 8 observation systems a-h. At the same time, we have modified the altitude ratio. The revised figure is as follows:

[Figure]

**34. Subfigures a to h are too small to read anything! On subfigure d I would expect the river to "flow in" on the "bottom" and "flow out" on the top according to the bigger map. However, the river is only half way through the map. Subfigure g is random and not understandable. You never mention**

**subfigure g in the text.**

Response: We have improved the clarity of sub-pictures a-h. In sub-figure d, the flow of the river is from the bottom to the top. Because sub-figure d is the Minqin soil observation system, our sampling points are concentrated in the middle and lower reaches of the Shiyang River, so only the middle and lower reaches of the Shiyang River are included in the sub-figure d. Subfigure g is a farmland observation system in Datan Township. Here we set up an experimental site to study the impact of human activities on agricultural production management. For example, we used hydrogen and oxygen stable isotopes to study the effect of mulching on soil moisture migration in farmland in arid oasis areas. We also added a description of each system in Section 3.1.

[Figure]

35. **Table 1: the table is missing an appropriate caption. As well, the units are not clear. i. e. precipitation mm/ year? Additionally, in the study site description you mention annual precipitation of max. 600mm, however, according to your table**

**one site received 1040mm (I guess per year). Same for temperature: air or water temperature?**

Response: We have added the title of Table 1, which is "Table 1 List of basic parameters". We revised all the units in Table 1. We modified the temperature to the annual average air temperature (℃) and the precipitation to the annual average precipitation (mm). We have also added the abbreviation of the sampling type and modified some of the information in Table 1. The revised table one is as follows:

Table 1 List of basic parameters

| Abbreviation | Full name | Longitude | Latitude | Elevation (m) | Average annual air temperature (℃) | Average annual precipitation ( mm) | Sampling type (abbreviation) | Sampling type (full name) | Location |
|---|---|---|---|---|---|---|---|---|---|
| QHLYXM | Qinghai Forestry Project | 101°51' | 37°32' | 3899 | - | - | hs | river water | a |
| MK | Colliery | 101°51' | 37°33' | 3647 | -0.20 | 595.10 | hs | precipitation | a |
| LXWL | Winding Road | 101°50' | 37°34' | 3305 | - | - | hs | river water | a |
| SDHHC | Tunnel Junction | 101°50' | 37°34' | 3448 | - | - | hs | river water | a |
| BDZ | Transformer Substation | 101°51' | 37°33' | 3637 | - | - | tr, zw, hs | soil, plant, river water | a |
| NQ | Ningqian | 101°49 | 37°37' | 3235 | - | - | hs | river water | a |
| SCG | Ningtanhe Middle East branch mixed water | 101°50' | 37°38' | 3068 | - | - | hs, js, tr | river water, precipitation, soil | a |
| MTQ | Wood Bridge | 101°53' | 37°41' | 2741 | - | - | hs | river water | a |
| SCLK | Three-way Intersection | 101°55' | 37°43' | 2590 | - | - | hs | river water | a |
| JTL | Nine Ridge | 102°02' | 37°51' | 2267 | - | - | dxs | groundwater | a |
| WGQ | The Bridge of the Cultural Revolution | 102°07' | 37°53' | 2174 | - | - | hs | river water | a |
| XYSK | Xiying Reservoir | 102°12' | 37°54' | 2058 | - | - | hs | river water | c |
| XYZ | Xiying Town | 102°26' | 37°58' | 1748 | 10.44 | 491.35 | js | precipitation | c |
| GGKFQ | Reform and Opening Bridge | 101°58' | 37°46' | 2590 | - | - | hs | river water | c |
| HJX | Huajian Township | 102°00' | 37°50' | 2390 | 7.65 | 262.64 | hs, dxs, js, tr | river water, groundwater, precipitation, | c |

| | | | | | | | | | |
|---|---|---|---|---|---|---|---|---|---|
| | | | | | | | | soil | |
| WW | Wuwei | 102°37' | 37°53' | 1581 | 5.23 | 300.14 | hs | river water | c |
| HLZ | Ranger Stations | 101°53' | 37°41' | 2721 | 3.24 | 469.44 | hs, js, tr, zw, dxs | river water, precipitation, soil, plant, groundwater | a |
| LLL | Lenglong Ridge | 101°28' | 37°41' | 3500 | 5.78 | 350.34 | js | precipitation | a |
| ZZXL | Zhuaxi Xiulong | 103°20' | 37°18' | 3556 | -2.37 | 500.17 | js | precipitation | d |
| JDT | Jiudun Beach | 102°45' | 38°07' | 1464 | 10.54 | - | js | precipitation | d |
| SCG | Shangchigou | 102°25' | 38°03' | 2400 | 7.28 | 377.13 | js, hs, dxs | precipitation, river water, groundwater | d |
| WWPD | Wuwei Basin | 102°42' | 38°06' | 1467 | - | - | js, dxs, tr, zw | precipitation, groundwater, soil, plant | d |
| DT | Dongtan | 102°47' | 38°16' | 1434 | 8.90 | 240.05 | hs,tr, zw | river water, soil, plant | e |
| HYSSK | Hongyashan Reservior | 102°53' | 38°24' | 1416 | 7.81 | 100.17 | hs, dxs, tr | river water, groundwater, soil | f |
| CQQ | Caiqi Bridge | 102°45' | 38°13' | 1443 | 5.63 | 300.26 | dxs, hs, tr, zw | groundwater, river water, soil, plant | d |
| XJG | Xiajiangou | 102°42' | 38°07' | 1200 | 9.36 | 110.18 | dxs | groundwater | d |
| HGG | Hongqi Valley | 102°50' | 38°21' | 1421 | 8.34 | 113.16 | js, dxs | precipitation, groundwater | d |
| BHZ | Protection Station | 102°29' | 38°09' | 2787 | - | - | dxs | groundwater | d |
| BDC | Beidong Township | 103°02' | 38°32' | 1367 | 9.52 | 155.45 | dxs | groundwater | g |
| XXWGZ | Xiyin Wugou Township | 102°58' | 38°29' | 1393 | - | - | dxs | groundwater | d |
| MQBQ | Minqin Dam | 103°08' | 39°02' | 1400 | 8.33 | 113.19 | tr | soil | d |
| QTH | Qingtu Lake | 103°36' | 39°03' | 1313 | 7.86 | 110.79 | js, dxs, ls, tr | precipitation, groundwater, lake water, soil | h |
| SWX | Suwu Township | 103°05' | 38°36' | 1372 | 9.82 | 155.84 | dxs, tr, zw, hs | groundwater, soil, plant, river water | d |
| DTX | Datan Township | 103°14' | 38°46' | 1349 | 11.49 | - | js, dxs, soi, zw, hs | precipitation, groundwater, soi, plant, river water | g |

| | | | | | | | | | |
|---|---|---|---|---|---|---|---|---|---|
| YXB | Yangxia Dam | 102°41' | 38°01' | 1489 | 10.76 | - | js, dxs, tr, zw | precipitation, groundwater, soil, plant | d |
| XBZ | Xuebai Toen | 103°01 | 38°32' | 1387 | 10.77 | - | js | precipitation | b |
| SYQ | Laboratory Area | 102°22' | 37°42' | 2438 | - | - | hs, tr | river water, soil | b |
| XCL | Small Valley | 102°24' | 37°43' | 2267 | - | - | hs | river water | b |
| JCLK | Intersection | 102°20' | 37°41' | 2544 | - | - | hs, tr | river water, soil | b |
| QSHSY | Spring River | 102°22' | 37°38' | 2747 | - | - | qs | spring water | b |
| HLD | Confluence | 102°26' | 37°44 | 2146 | - | - | hs, tr, zw | river water, soil, plant | b |
| QXZ | Meteorological Station | 102°20' | 37°42 | 2543 | 3.34 | 510.56 | js, dxs | precipitation, groundwater | b |
| BGH | Binggou River | 102°17' | 37°40' | 2872 | 5.28 | - | hs, tr, | river water, soil | b |
| NCHHLH | South Nancha River | 102°26' | 37°43' | 2163 | - | - | hs | river water | b |
| LKS | Two Pine | 102°17' | 37°40' | 2832 | 5.69 | - | hs, tr | river water, soil | b |
| NYSKRK | Nanying Reservoir | 102°29' | 37°47' | 1955 | 7.82 | 330.16 | hs | river water | b |
| SGZZ | Sigou stckade | 102°23' | 37°40' | 2492 | 10.34 | 675.54 | hs | river water | b |
| JZGD | Construction Site | 102°25' | 37°41' | 2303 | - | - | hs | river water | b |
| QLX | Qilian Township | 102°42' | 38°08' | 3394 | 5.13 | 300.15 | js, qs | precipitation, spring water | d |
| XYWG | Xiying Wugou | 102°10' | 37°53' | 2097 | 7.99 | 197.67 | hs, js, tr, zw | river water, precipitation, soil, plant | c |
| HSH | Hongshui River | 102°45' | 38°13' | 1454' | - | - | hs | river water | d |
| XCL | Small village | 102°24' | 37°43' | 2267 | - | - | hs | river water | b |
| YHRJ | A family | 102°20' | 37°42' | 2543 | - | - | hs | river water | b |

**36. Figure 3: which water bodies / soil water sites are presented here?**

Response: Figure 3 shows the isotope values of all samples collected in the Shiyang River Basin from 2015 to 2019. The precipitation isotope values come from 19 sampling points in the Shiyang River Basin, and the surface water isotope values come from 34 sampling points in the Shiyang River Basin. The groundwater isotope values come from 17 sampling points in the Shiyang River Basin, and the plant water isotope values are from 12 sampling points in the Shiyang River Basin.

---

## Author Comment (AC2)

Dear Reviewer:

Thank you for your letter and for the reviewer's comments concerning our manuscript entitled "Stable water isotope monitoring network of different water bodies in Shiyang River basin, a typical arid river in China" (Manuscript Number: essd-2021-79).

According to the reviewer's comments, we have carefully checked the data and research methods, and seriously modify our manuscript. The modified portions have been marked in red in the manuscript changes version. The primary corrections and the response to the reviewers' comments are as follows.

**Responses to the reviewer's comments:**

In the previous 30 years, I have done a lot of similar work in the European Alps and Asian Hengduan Mountains, but I have to say that this is an impressive article. The author measured the stable isotopes of different water bodies in the whole Shiyang River Basin and collected corresponding hydrological and meteorological data in this manuscript. The data set from 2015 to 2019, with 53 observation points and 6760 experimental data obtained. This data set is one of the most systematic data sets (the matching of different water bodies is almost perfect) I have seen so far, and it is very influential, the key is that the data are all sample experimental data. I checked the data set, and the observation is very systematic and scientific (at least in the various documents I have seen ). I believe that the publication of this data set will promote the research of global isotope hydrology. Therefore, I support the publication of this article as soon as possible! However, a significant article must have good writing, and this article needs further improvement in expression and language.

Response: We have gradually improved it by carefully revising every recommendation you mentioned.

**Major comments**

1. Data articles should be easy to read and use by other researchers, the entire manuscript, including the data set, lacks some basic information, especially information about experiments and sample collection. The six observation systems established by the author are excellent and should be illustrated the purpose of observation, you cannot expect that every reader is a professional, and the writing should be clear.

Response: Thank you very much for your suggestion. We have added information from experiments and sample collection to the manuscript and the added content is as follows:

**4 Data and Methods**

**4.1 Sample collection**

**4.1.1 Collection of precipitation**

We have set up 16 weather stations in the Shiyang River Basin to collect precipitation, including rain barrels used to collect precipitation. The rain barrel is placed in an open outdoor area and consists of rain gear, funnel, water bottle and rain cup. The diameter of the rain gear is 20 cm, and the port of the device is horizontal. The height of the rain opening of the instrument is set to 70 cm from the ground level. We placed an anti-evaporation polyethene ball at the funnel's mouth and added a layer of paraffin oil to the bottom of the container to prevent evaporation from causing isotope fractionation. Immediately after each precipitation event, transfer the collected liquid precipitation to a 100 ml high-density sample bottle. Solid precipitation must be transferred to a high-density polyethene sample bottle after it becomes liquid water at room temperature (23°C). The sample bottle is sealed with Parafilm. The polyethene bottle is labelled simultaneously, indicating the date of collecting the sample, the type of precipitation (rain, snow, hail), and the volume of precipitation. Store the collected samples in a refrigerator at about 4°C for later analysis. For multiple precipitation events in a day, we sample by precipitation events.

**4.1.2 Collection of surface water and groundwater**

Polyethene bottles are used to collect surface water (rivers, lakes, reservoirs). Stratified sampling is carried out at different depths (surface layer, middle layer, bottom layer. Groundwater samples were obtained from the groundwater monitoring wells of the Shiyang River Basin Administration, China Hydrological Administration and Gansu Hydrological Administration. The bottle of the sample is sealed with parafilm film and then frozen until the experiment. Simultaneously, the polyethene bottle sample is labelled with the sample's date, sampling depth, and the stream and tributary stream. The collected water samples should be placed in places where the sunlight is not direct to avoid the evaporation of water, which would affect the validity of the data. The samples were taken back to the freezer in the laboratory for refrigeration within 10 hours.

**4.1.3 Collection of soil and plant water**

The soil sample is collected at a depth of 100cm, and samples are taken sequentially at 10cm intervals. The upper reaches of the Shiyang River Basin are mainly clay, and the middle and lower reaches are clay and sand, but sand is the main soil type. Table 2 shows the characteristics of soil in the farmland area of the Shiyang River Basin The soil samples collected were divided into two parts, put into a 50 ml glass bottle. The bottle mouth was sealed with parafilm membrane and transported to the observation station within 10 hours after the sampling date was marked for cryopreservation to test stable isotope data. The other part of the sample was placed in a 50 ml aluminium box and used the drying method to test the soil moisture content since 2019.

| Soil depth (cm) | Clay (%) | Silt (%) | Sand (%) | Soil bulk density (g/cm 3 ) |
|-----------------|----------|----------|----------|----------------------------------------|
| 0-10            | 10.20    | 38.85    | 50.95    | 1.052                                  |
| 10-20           | 12.94    | 37.76    | 49.30    | 1.194                                  |
| 20-30           | 10.33    | 44.23    | 45.44    | 1.298                                  |
| 30-40           | 13.48    | 38.69    | 47.83    | 1.180                                  |
| 40-50           | 12.01    | 35.09    | 52.90    | 1.140                                  |
| 50-60           | 11.21    | 42.83    | 45.96    | 1.206                                  |
| 60-70           | 10.34    | 42.98    | 46.68    | 1.208                                  |
| 70-80           | 11.09    | 38.96    | 49.95    | 1.106                                  |
| 80-90           | 11.75    | 37.72    | 50.53    | 1.200                                  |
| 90-100          | 7.21     | 35.97    | 56.82    | 1.272                                  |

 Table 2 Basic information of soil samples
 (Zhu et al., 2021)

Plant sample collection: For trees and shrubs, we collect xylem, and for herbs,

we collect non-green parts at the junction of rhizomes. When sampling, we use scissors to collect vegetation stems, peel off the bark, put them in 50ml glass bottles, seal them, and freeze them until experimental analysis. Table 3 shows the plant information collected in the Shiyang River Basin.

| Sampling points | Vegetation types                                                | Sample size |
|-----------------|-----------------------------------------------------------------|-------------|
| BDZ             | Agropyron cristatum                                             | 30g ±0.5    |
| CQQ             | Corn (stem), reed, jujube (Branches), dryland willow (Branches) | 30g-100g    |
| DT              | Reed                                                            | 30g±0.5     |
| DTX             | Spring wheat (stem), corn (root, stem)                          | 30g±0.5     |
| HJX             | Willow (Branches)                                               | 100g±0.5    |
| HLD             | Qinghai Spruce (Branches)                                       | 100g±0.5    |
| HLZ             | Qinghai Spruce (Branches)                                       | 100g±0.5    |
| WWPD            | Corn (stem), wheat (stem)                                       | 30g±0.5     |
| XYWG            | Poplar (Branches), wheat                                        | 100g±0.5    |
| YXB             | Corn (stem)                                                     | 30g±0.5     |
| SWX             | Corn, wheat (stem)                                              | 30g±0.5     |
| LLL             | Salsola purpurea                                                | 30g±0.5     |

**Table 3 Basic information of plant samples**

**4.1.4 Collection of meteorological data**

The local meteorological data were obtained and recorded during the sampling period by the automatic weather stations (watchdog 2000 series weather stations) erected near the sample plot. Meteorological data include temperature ( $^{\circ}$ C), relative humidity (%), atmospheric pressure (hPa), dew point temperature ( $^{\circ}$ C) and precipitation amount (mm).

**4.2 Experiment analysis**

**4.2.1 Water extraction experiment**

We use vacuum condensation to extract soil and plant water. The extraction equipment used is LI-2100 automatic vacuum condensation extraction equipment. Before water extraction is performed on the soil and plants, the collected samples need to be taken out of the refrigerator to thaw, and each sample bottle should be stuffed with a small ball of cotton to prevent the water from evaporating. When extracting water, we set the extraction time to 150 minutes (180 minutes for plants), the temperature to 190°C, the upper limit of the vacuum pressure to 800pa, and the leakage rate to 0. The water evaporates from the soil or plant sample by heating it for

a specified time and then freezes it in a liquid nitrogen cold trap. After the extraction is completed, the sample is thawed at room temperature and shaken. Use a 1ml syringe to extract the water sample into a labelled sample bottle, seal it and wait for the isotope experiment.

**4.2.2 Isotope experiment**

All the collected water samples were analyzed in the stable isotope laboratory of Northwest Normal University using liquid water isotope analysis (DLT-100, Los Gatos Research, USA). Each water sample and isotope standard sample were injected six times in a row. We discarded the first two injections and used the average of the last four times as the final result to eliminate the instrument memory effect. The result of the isotope measurement is expressed by the symbol " $\delta$ " and expressed in thousandths of the difference relative to the Vienna Standard Mean Ocean Water (Craig, 1961):

$$\delta_{sample} (\%_{0}) = \left[ \left( \frac{R_{sample}}{R_{v-smow}} \right) - 1 \right] \times 1000$$
(1)

In the formula, Rsample is the ratio of 18O/16O or 2H/1H in the collected sample, Rv-smow is the ratio of 18O/16O or 2H/1H in the Vienna standard sample.The analytical accuracy of  $\delta^{2}$ H and  $\delta^{18}$ O are ±0.6‰ and ±0.2‰, respectively.

**4.2.3 Modification of plant water isotope data**

Suppose the water sample contains compounds with the same absorption characteristics of the same wavelength. In that case, it will lead to errors in the measurement of the laser liquid water analyzer, and the most likely pollutants to cause errors are methanol and ethanol. So using deionized water with different concentrations of pure methanol and ethanol, the combination of Los Gatos company LWIA - Spectral Contamination Identifier v1.0 Spectral analysis software (NB) to determine methanol and ethanol (BB) pollution degree of spectrum measurement, establishing the  $\delta^2$ H and  $\delta^{18}$ O correction method for the spectra of pollution (Meng et al., 2012; Liu et al., 2013). In the correction process, methanol and ethanol solution concentration configuration were similar to Meng's experiment (2012). Correction

results for methanol its broadband measurements of NB metric logarithmic respectively with  $\Delta\delta D$  and  $\Delta\delta^{18}O$  are significantly quadratic curve relationship, respectively is:

$$\Delta \delta D = 0.018 (\ln NB)^3 + 0.092 (\ln NB)^2 + 0.388 \ln NB + 0.785 (R^2 = 0.991, p > 0.0001)$$
(2-1)

$$\Delta \delta^{18} O = 0.017 (\ln NB)^3 - 0.017 (\ln NB)^2 + 0.545 \ln NB + 1.356 (R^2 = 0.998, p < 0.0001)$$
(2-2)

Its broadband measurements for ethanol correction results in BB metric and  $\Delta\delta D$ and  $\Delta\delta^{18}O$ , a quadratic curve and linear relationship respectively, are:

$$\Delta \delta D = -85.67BB + 93.664(R^2 = 0.747, p = 0.026)(BB < 1.2)$$
(2-3)

$$\Delta \delta^{18} O = -21.421BB^2 + 39.935BB - 19.089(R^2 = 0.769, p < 0.012)$$
(2-4)

**4.3 Data quality**

It has always been a difficult problem to control the experimental error to the minimum. We use Manner-Kendall to test meteorological and hydrological data, eliminate abnormal values, and use interpolation to obtain vacant values. For the isotope data, we first use the LIMA software to check the original isotope data. We stipulate that when one or more of the 6 data of a sample is marked in red, we call the sample an error sample, even though the sample is in the display on the liquid water analyzer is normal, and we will re-experiment the sample until it passes the LIMA software verification. Then we will use SPSS software to check the normality of the obtained isotope data. At present, our errors mainly come from the following aspects:

**4.3.1 Sample collection**

The error caused by precipitation sample collection mainly comes from the

sampling personnel's failure to transfer the rainwater in the rain gauge or mixing two or more rainwaters after each precipitation event, which will cause the rainwater in the rain gauge to evaporate and affect the data.

The error is caused by the collection of vegetation samples mainly from the samples collection process, and the vegetation is exposed to the air, which causes the vegetation to fractionate water.

The error in collecting soil samples is that we collected soil samples that contained many microorganisms, which impact the data results.

**4.3. 2 Experiment**

The experimental error is mainly because we set the same moisture extraction parameters for samples with different soil characteristics. It is difficult to make post-mortem corrections for soil properties or the effects of extraction conditions because such information is rarely reported, and massive variability in method details is common (Walker et al., 1994). In addition, there are still measurement uncertainties during the extraction of water, which also come from the loss of water vapor during the vacuum of the extraction system and the non-temperature heating temperature, which will lead to experimental errors.

Our calibration of plant sample data only considers methanol and ethanol pollution, but the plant and soil water extracts may contain various other pollutants, leading to experimental errors. In addition, studies have shown that the mismatch between xylem and plant water sources is due to the fractionation of isotopes in the process of water absorption (Poca et al., 2019), which questioned the fact that plants do not undergo fractionation during the process of water absorption (Porporato, 2001; Meissne et al., 2014) this traditional view. However, there is no better solution, so we still use traditional methods to collect samples and conduct experiments.

2. The author has been publishing data continuously, and there are many other data sets worldwide. The compatibility and matching of data should be considered. Therefore, the author should add isotope experiments in the current manuscript, especially the reference standards for isotope data, which is very important for data quality.

Response: Thank you very much for your suggestion. We have added reference standards for isotope experiments and isotope data to the manuscript, and the added content is as follows:

**4.2.2 Isotope experiment**

All the collected water samples were analyzed in the stable isotope laboratory of Northwest Normal University using liquid water isotope analysis (DLT-100, Los Gatos Research, USA). Each water sample and isotope standard sample were injected six times in a row. We discarded the first two injections and used the average of the last four times as the final result to eliminate the instrument memory effect. The result of the isotope measurement is expressed by the symbol " $\delta$ " and expressed in thousandths of the difference relative to the Vienna Standard Mean Ocean Water (Craig, 1961):

$$\delta_{sample} (\%_0) = \left[ \left( \frac{R_{sample}}{R_{v-smow}} \right) - 1 \right] \times 1000$$
(1)

In the formula, Rsample is the ratio of 18O/16O or 2H/1H in the collected sample, Rv-smow is the ratio of 18O/16O or 2H/1H in the Vienna standard sample.The analytical accuracy of  $\delta^2$ H and  $\delta^{18}$ O are ±0.6‰ and ±0.2‰, respectively.

3. A good author should think critically about the problem. In the current version of the manuscript, I have not seen the author's critical comments on stable isotope technology.

Response: We have added a critical comment on stable isotope technology to the manuscript, adding the following content:

Simultaneously, the fractionation of isotopes also runs through every link of the water cycle (Song et al., 2007; Dansgaard, 1953, 1964). For example, Meißner et al. (2014) emphasized that the change of  $\delta^{18}$ O largely depends on the soil type (Araguás-Araguás et al., 1995). Orlowskii et al. (2016) showed that incomplete water extraction in the cryogenic distillation process might fractionate water isotopes. Therefore, clay requires a longer extraction time and temperature to reduce the fractionation effect in the extraction process. In addition, a study by Sofer and Gat in 1975 showed that the formation of hydrated spheres around cations in aqueous solutions would fractionate the oxygen isotopes of the water. Gaj et al. (2017) showed that the isotope characteristics are biased due to a process different from Rayleigh distillation that we cannot reduce the effect caused by the mineral-water interaction entirely.

Studies have shown that physicochemical soil properties may cause the fractionation of hydrogen and oxygen in soil water (Meißner et al., 2014). Because we do not know whether the unstable hydrogen fraction during the low-temperature extraction process will cause isotope fractionation (Orlowski et al., 2016). In addition, we know little about the effect of soil microbial activity on the extracted water isotope results (Orlowski et al., 2018). However, from previous studies, it is still difficult to make post-correction in terms of soil properties or the effects of extraction conditions because such information is rarely reported, and huge variability in method details is common (Walker et al., 1994). We have always known these"problems" exist, but water vacuum extraction is still the standard method for extracting soil and plant water in ecological hydrology (Ingraham et al., 1992). In most plants, the isotopic composition of water does not change due to root absorption and transport through the stem xylem (White et al., 1985). However, more and more studies have shown a difference between the isotope composition of xylem water and plant water sources (Poca et al., 2019), and the fractionation can occur along the root water absorption pathway. This fact does not make the isotope method in the soil-plant-atmosphere useless to track water in the continuum, and  $\delta D$  and  $\delta^{18}O$  can still be used similarly to study water absorption of various plants (Poca et al., 2019).

- Araguás-Araguás, L., Rozanski, K., Gonfifiantini, R., and Louvat, D.: Isotope effects accompanying vacuum extraction of soil water for stable isotope analyses, Journal of Hydrolog, 168, 159–171, https://doi.org/10.1016/0022-1694(94)02636-P, 1995.
- Orlowski, N., Breuer, L., and McDonnell, J. J.: Critical issues with cryogenic extraction of soil water for stable isotope analysis. Ecohydrology, 9(1) 1-5, 2016.
- Sofer, Z., and Gat, J.: The isotope composition of evaporating brines: effect of the isotopic activity ratio in saline solutions. Earth Planet. Sci. Lett. 26 (2), 179–186, 1975.
- Gaj, M., Kaufhold, S., Königer, P., Beyer, M., and Himmelsbach, T.: Mineral mediated isotope fractionation of soil water. Rapid Communications in Mass Spectrometry, 31(3), https://doi.org/10.1002/rcm.7787, 2017.
- Meißner, M., Köhler, M., Schwendenmann, L., Hölscher, D., Dyckmans, J.: Soil water uptake by trees using water stable isotopes (d2H and d18O) a method test regarding soil moisture, texture and carbonate. Plant and Soil 376 (1–2), 327–335, https://doi.org/10.1007/s11104-013-1970-z, 2014.
- Orlowski, N., Breuer, L., Angeli, N., Boeckx, P., and Mcdonnell, J. J.: Inter-laboratory comparison of cryogenic water extraction systems for stable isotope analysis of soil water. Hydrology and Earth System Sciences Discussions, 1-36. https://doi.org/10.5194/hess-2018-128, 2018.
- Walker, G. R., Woods, P. H., and Allison, G. B.: Interlaboratory comparison of methods to determine the stable isotope composition of soil water, Chem. Geol., 111, 297–306, https://doi.org/10.1016/0009-2541(94)90096-5, 1994.
- Walker, G. R., Woods, P. H., and Allison, G. B.: Interlaboratory comparison of methods to determine the stable isotope composition of soil water, Chem. Geol., 111, 297–306, https://doi.org/10.1016/0009-2541(94)90096-5, 1994.
- Poca, M., Coomans, O., Urcelay, C., Zeballos, S. R , and Boeckx, P.: Isotope fractionation during root water uptake by acacia caven is enhanced by arbuscular

mycorrhizas. Plant and Soil (3), https://doi.org/10.1007/s11104-019-04139-1, 2019.

Ingraham, N. L., and Shadel, C.: A comparison of the toluene distillation and vacuum/heat methods for extracting soil water for stable isotopic analysis. Journal of Hydrology, 140(1), 371-387, https://doi.org/10.1016/0022-1694(92)90249-U, 1992.

4. In addition to providing rich data to readers, data articles should also guide readers to use these data to solve scientific problems. In the current version of the manuscript, the author's outlook on the data set is short, and it is difficult for readers to be inspired by this article.

Response: We have added relevant content to the summary and outlook of the manuscript. The added content is as follows:

This study collected the stable isotope values of different water bodies in the Shiyang River Basin of China's arid inland river basin from since 2019. As of 2019, we have collected 6,760 isotope data one by one and formed a data set. Through the analysis of different water stable isotopes in the Shiyang River Basin, the main conclusions reached are: (1) The slope and intercept of the LMWL in the Shiyang River Basin are both smaller than GMWL, which is the same as that of the study area located in the arid inland of Northwest China. When raindrops fall to the surface, they are subject to strong evaporation. (2) The main source of replenishment in the Shiyang River Basin is precipitation. The stable isotopes in the Shiyang River Basin show periodic fluctuations, depleted in winter and spring and enriched in summer and autumn. (3) The fluctuation of the stable isotope of lake water is greater than that of other water bodies, and the coefficient of variation is the largest. This is related to the large seasonal differences in the evaporation. (4) The change of groundwater lags behind that of surface water. This data set provides a new basis for studying the stable water isotopes of different water bodies in the inland river basins of China. Through these data, we can study the water conservancy connection between different water bodies, study the impact of human activities on water cycle, and provide certain guidelines for the rational use of water resources in arid regions.

The accuracy of the isotope data we obtained is relatively high, but we have ignored some factors that lead to experimental errors in the experiment. For example, we have soil microorganisms in the collected soil samples, and the activity of soil microorganisms may affect the experiment. During the extraction experiment, we did not consider the nature of the soil and set the experimental parameters of all soil samples to be the same. However, the eco-hydrology and soil science communities still lack standard norms and suggestions on these issues. Studies have shown that new continuous in situ measurements of the isotopic composition of soil and plant water may overcome the problem of isotopic fractionation observed when we extract water. However, the fractionation problem caused by different soil characteristics during the experiment can only be studied through a large number of experiments in the future.

As our field observations proceed, our data set will be updated year by year. To improve the dataset, we encourage users of the dataset to contact the author for suggestions.

5. As a data set article, there are many soil and vegetation data in the data set. Among them, there are 3,779 soil samples and 509 plant samples. The acquisition of these data is essential. I believe this is also the study of agricultural activities and crop water use in the arid regions of Central Asia. However, the introductory part of the article focuses on the indications of stable isotopes of precipitation to the water cycle. It is recommended that the relevant discussions on isotope ecology be added to the introductory part of the article.

Response: We have added the relevant content of isotope ecology to the manuscript, and the added content is as follows:

Most surface water comes directly from runoff from rainfall, groundwater discharge, or a combination of these two water bodies (Surinaidu et al.,2012; Hutchinset al., 2018). Once surface water is exposed to rivers or lakes, it may change the isotope values of surface water through evaporative fractionation (Gremillion et al., 2000; Andrew et al., 1992). The lake has more time for direct contact with the atmosphere due to its long residence time, making lakes more susceptible to

evaporation than rivers, leading to isotope changes (Ambrosetti et al., 2003). Because surface water easily changes its isotope value through evaporation, the observation of surface water is usually regarded as limited in practice (Gat and Airey, 2006). Generally speaking, we measure surface water for a specific purpose, such as finding local leak sources (for example, from a dam) (Zhu et al., 2021) or determining the larger-scale hydraulic connection between surface water and groundwater (Atkinson et al., 2015). In the farmland ecosystem, whether it is atmospheric precipitation or irrigation water, it can only provide the water needed for crop growth after it is converted into soil water (Zhu et al., 2021). Soil water is the centre of mutual transformation of atmospheric precipitation, surface water, groundwater and plant water and has an essential impact on the regional water cycle (Liu et al., 1997). Although technology development has made it easier for us to analyze stable isotopes in soil water, the fractionation mechanism of soil water isotope is affected by factors such as local climate, soil texture, hydrological conditions and human activities. Therefore, more research is needed to accurately understand the soil water activity of farmland and the law of crop water absorption.

- Ambrosetti, W., Barbanti, L., and Sala, N.: Residence time and physical processes in lakes. Journal of Limnology, 62(1), 1-15. https://doi.org/10.4081/jlimnol.2003.s1.1, 2003.
- Andrew, H. L., Chris, B. J., Phillip, M. G., and John, O. M.: A stable isotope investigation of groundwater-surface water interactions at lake tyrrell, victoria, australia. Chemical Geology, 96(1-2), 19-32, https://doi.org/10.1016/0009-2541(92)90119-P, 1992.
- Atkinson, A. P., Cartwright, I., Gilfedder, B. S., Hofmann, H., Unland, N. P., Cendón,
  D. I., and Chisari, R.:A multi-tracer approach to quantifying groundwater inflows to an upland river; assessing the influence of variable groundwater chemistry. Hydrological Processes, https://doi.org/10.1002/hyp.10122, 2015.
- Gat, J. R., and Airey, P. L.: Stable water isotopes in the atmosphere/biosphere/lithosphere interface: scaling-up from the local to

continental scale, under humid and dry conditions. Global and Planetary Change, 51(1/2), 25-33, https://doi.org/10.1016/j.gloplacha.2005.12.004, 2006.

- Gremillion, P., and Wanielista, M.: Effects of evaporative enrichment on the stable isotope hydrology of a central florida (USA) river. Hydrological Processes, 14(8), 1465-1484, 2000.
- Hutchins, M. G., Abesser, C., Prudhomme, C., Elliott, J. A., Bloomfield, J. P., Mansour, M. M., and Hitt. O, E.: Combined impacts of future land-use and climate stressors on water resources and quality in groundwater and surface waterbodies of the upper Thames river basin, UK. Science of the Total Environment, 631-632, 962, https://doi.org/10.1016/j.scitotenv.2018.03.052, 2018.
- Liu, C. M.: Research on the interface process of water movement in the soil-plant-atmosphere system. Acta Geographica Sinica, 64(4), 366-373, https://doi.org/10.11821/xb199704011, 1997.
- Surinaidu, L., Bacon, C., and Pavelic, P.: Agricultural groundwater management in the upper bhima basin, India: current status and future scenarios. Hydrology and Earth System Sciences, 9(9), https://doi.org/ 10.5194/hess-17-507-2013, 2012.
- Zhu, G. F., Sang, L.Y., Zhang, Z. X., Sun, Z. G., Ma, H. Y., Liu, Y. W., Zhao, K. L.,
  Wang, L., and Guo Huiwen.: Impact of landscape dams on river water cycle in urban and peri-urban areas in the Shiyang River Basin: Evidence obtained from hydrogen and oxygen isotopes: Journal of Hydrology, https://doi.org/10.1016/J.JHYDROL.2021.126779, 2021.
- Zhu, G. F., Yong, L. L., Zhang, Z. X., Sun, Z. G., Wan, Q. Z., Xu, Y. X., Ma, H. Y., Sang, L. Y., Liu, Yu. W., Wang, L., Zhao, K. K., and Guo, H.W.: Effects of plastic mulch on soil water migration in arid oasis farmland: Evidence of stable isotopes: Catena, https://doi.org/10.1016/J.CATENA.2021.105580, 2021.

**Specific comments:**

1. L11-12: I think time information should be added here.

Response: We have added the time information, the revised content is as follows:

We have established a stable water isotope monitoring network since 2015 in the Shiyang River Basin in China'arid northwest.

2. L15-16: Arrange six observation systems in the order of upstream, midstream, and downstream.

Response: We have adjusted the order of these six observing systems according to your suggestion, and the revised content is as follows:

The monitoring station covers the upper, middle and lower reaches of the river basin, with six observation systems: river source area, oasis area, ecological restoration area, reservoir canal system area, oasis farmland area, and salinized area.

3. L21: Change stable isotope data to water stable isotope data, the same in other parts of the manuscript, please keep the terminology consistent in the manuscript.

Response: We have made changes based on your suggestions, and the revised content is as follows:

The data set includes stable water isotope data, meteorological data and hydrological data in the Shiyang River Basin.

4. L24: "these " not "theae ".

Response: We have made changes based on your suggestions, and the revised content is as follows:

This observation network's construction provides us with stable water isotopes data and hydrometeorological data, and we can use these data for hydrological and meteorological related scientific research.

5. L26: How to provide a scientific basis for the construction of water conservancy projects in arid areas? The author did not mention in the manuscript.

Response: We have added a description of the purpose of setting up six observation systems in the manuscript. Among them is a mention of the impact of water conservancy projects on arid areas: we used stable isotopes to analyze the effect of reservoirs on plant water use strategies and water conservancy projects on the water cycle in arid regions. We studied the isotope characteristics of different water bodies in cities and suburbs in the upper and middle reaches of the Shiyang River Basin from 2015 to 2019 and assessed the hydrological effects of urban landscape dams at the

basin scale in arid regions where water resources are scarce and ecosystems are fragile. Our results of evaporation loss rate show that landscape water is 0-5% higher. Moreover, the cumulative effect of multiple landscape dams has led to a large loss of water resources in arid regions. The study also shows that evaporation is an essential factor leading to changes in the isotope composition of landscape water in the Shiyang River Basin. Therefore, we believe that the potential adverse effects of urban landscape dams in arid regions should be highly considered in long-term water sustainability planning (Zhu et al., 2021).

Zhu, G. F., Sang, L.Y., Zhang, Z. X., Sun, Z. G., Ma, H. Y., Liu, Y. W., Zhao, K. L., Wang, L., and Guo Huiwen.: Impact of landscape dams on river water cycle in urban and peri-urban areas in the Shiyang River Basin: Evidence obtained from hydrogen and oxygen isotopes: *J*ournal of Hydrology, https://doi.org/10.1016/J.JHYDROL.2021.126779, 2021.

6. L37: The format of the references needs to be revised.

Response: We have revised the reference based on your comment, and the revised content is as follows:

Simultaneously, the fractionation of isotopes also runs through every link of the water cycle (Song et al., 2007; Dansgaard, 1953, 1964)

7. L41: "Hepp et al., 2015" not "Hepp et al. 2015", please pay attention to the punctuation in the manuscript.

Response: We have made changes based on your comments, and the revised content is as follows:

Stable isotopes of hydrogen and oxygen in water have been widely used in the water cycle (Gibson et al., 2010; Penna et al., 2013; Timsic et al., 2014; Evaristo et al., 2015; Negrel et al., 2016), paleoclimate and paleoenvironmental evolution (Wei et al., 1994; Speelman et al., 2010; Steinman et al., 2010; Hepp et al., 2015), reconstruction of pale plateau height (Thompson et al., 2000; Yao et al., 2008; Xu et al., 2015; Li et al., 2017) and other fields.

8. L49: I think adding the control factors of other water body isotopes here will be a good combination with the previous precipitation isotope factors.

Response: We have added control factors for other water bodies according to your suggestions, and the added content is as follows:

Most surface water comes directly from runoff from rainfall, groundwater discharge, or a combination of these two water bodies (Surinaidu et al.,2012; Hutchinset al., 2018). Once surface water is exposed to rivers or lakes, it may change the isotope values of surface water through evaporative fractionation (Gremillion et al., 2000; Andrew et al., 1992). The lake has more time for direct contact with the atmosphere due to its long residence time, making lakes more susceptible to evaporation than rivers, leading to isotope changes (Ambrosetti et al., 2003). Because surface water easily changes its isotope value through evaporation, the observation of surface water is usually regarded as limited in practice (Gat and Airey, 2006). Generally speaking, we measure surface water for a specific purpose, such as finding local leak sources (for example, from a dam) (Zhu et al., 2021) or determining the larger-scale hydraulic connection between surface water and groundwater (Atkinson et al., 2015). In the farmland ecosystem, whether it is atmospheric precipitation or irrigation water, it can only provide the water needed for crop growth after it is converted into soil water (Zhu et al., 2021). Soil water is the centre of mutual transformation of atmospheric precipitation, surface water, groundwater and plant water and has an essential impact on the regional water cycle (Liu et al., 1997). Although technology development has made it easier for us to analyze stable isotopes in soil water, the fractionation mechanism of soil water isotope is affected by factors such as local climate, soil texture, hydrological conditions and human activities. Therefore, more research is needed to accurately understand the soil water activity of farmland and the law of crop water absorption.

- Ambrosetti, W., Barbanti, L., and Sala, N.: Residence time and physical processes in lakes. Journal of Limnology, 62(1), 1-15. https://doi.org/10.4081/jlimnol.2003.s1.1, 2003.
- Andrew, H. L., Chris, B. J., Phillip, M. G., and John, O. M.: A stable isotope investigation of groundwater-surface water interactions at lake tyrrell, victoria, australia.
  Chemical Geology, 96(1-2), 19-32,

https://doi.org/10.1016/0009-2541(92)90119-P, 1992.

- Atkinson, A. P., Cartwright, I., Gilfedder, B. S., Hofmann, H., Unland, N. P., Cendón,
  D. I., and Chisari, R.:A multi-tracer approach to quantifying groundwater inflows to an upland river; assessing the influence of variable groundwater chemistry. Hydrological Processes, https://doi.org/10.1002/hyp.10122, 2015.
- Gat, J. R., and Airey, P. L.: Stable water isotopes in the atmosphere/biosphere/lithosphere interface: scaling-up from the local to continental scale, under humid and dry conditions. Global and Planetary Change, 51(1/2), 25-33, https://doi.org/10.1016/j.gloplacha.2005.12.004, 2006.
- Gremillion, P., and Wanielista, M.: Effects of evaporative enrichment on the stable isotope hydrology of a central florida (USA) river. Hydrological Processes, 14(8), 1465-1484, 2000.
- Hutchins, M. G., Abesser, C., Prudhomme, C., Elliott, J. A., Bloomfield, J. P., Mansour, M. M., and Hitt. O, E.: Combined impacts of future land-use and climate stressors on water resources and quality in groundwater and surface waterbodies of the upper Thames river basin, UK. Science of the Total Environment, 631-632, 962, https://doi.org/10.1016/j.scitotenv.2018.03.052, 2018.
- Liu, C. M.: Research on the interface process of water movement in the soil-plant-atmosphere system. Acta Geographica Sinica, 64(4), 366-373, https://doi.org/10.11821/xb199704011, 1997.
- Surinaidu, L., Bacon, C., and Pavelic, P.: Agricultural groundwater management in the upper bhima basin, India: current status and future scenarios. Hydrology and Earth System Sciences, 9(9), https://doi.org/ 10.5194/hess-17-507-2013, 2012.
- Zhu, G. F., Sang, L.Y., Zhang, Z. X., Sun, Z. G., Ma, H. Y., Liu, Y. W., Zhao, K. L.,
  Wang, L., and Guo Huiwen.: Impact of landscape dams on river water cycle in urban and peri-urban areas in the Shiyang River Basin: Evidence obtained from hydrogen and oxygen isotopes: Journal of Hydrology, https://doi.org/10.1016/J.JHYDROL.2021.126779, 2021.
- Zhu, G. F., Yong, L. L., Zhang, Z. X., Sun, Z. G., Wan, Q. Z., Xu, Y. X., Ma, H. Y.,

Sang, L. Y., Liu, Yu. W., Wang, L., Zhao, K. K., and Guo, H.W.: Effects of plastic mulch on soil water migration in arid oasis farmland: Evidence of stable isotopes: Catena, https://doi.org/10.1016/J.CATENA.2021.105580, 2021.

9. L62: Compared with traditional hydrological methods, what disadvantages are hydrogen and oxygen stable isotope technology? In the current manuscript, I have not seen critical comments on the stable isotopes of hydrogen and oxygen.

Response: According to your suggestion, we have added the disadvantages of hydrogen and oxygen stable isotope technology compared with traditional hydrological methods. The added content is as follows:

Simultaneously, the fractionation of isotopes also runs through every link of the water cycle (Song et al., 2007; Dansgaard, 1953, 1964). For example, Meißner et al. (2014) emphasized that the change of  $\delta^{18}$ O largely depends on the soil type (Araguás-Araguás et al., 1995). Orlowskii et al. (2016) showed that incomplete water extraction in the cryogenic distillation process might fractionate water isotopes. Therefore, clay requires a longer extraction time and temperature to reduce the fractionation effect in the extraction process. In addition, a study by Sofer and Gat in 1975 showed that the formation of hydrated spheres around cations in aqueous solutions would fractionate the oxygen isotopes of the water. Gaj et al. (2017) showed that the isotope characteristics are biased due to a process different from Rayleigh distillation that we cannot reduce the effect caused by the mineral-water interaction entirely.

Studies have shown that physicochemical soil properties may cause the fractionation of hydrogen and oxygen in soil water (Meißner et al., 2014). Because we do not know whether the unstable hydrogen fraction during the low-temperature extraction process will cause isotope fractionation (Orlowski et al., 2016). In addition, we know little about the effect of soil microbial activity on the extracted water isotope results (Orlowski et al., 2018). However, from previous studies, it is still difficult to make post-correction in terms of soil properties or the effects of extraction conditions because such information is rarely reported, and huge variability in method details is common (Walker et al., 1994). We have always known these"problems" exist, but

water vacuum extraction is still the standard method for extracting soil and plant water in ecological hydrology (Ingraham et al., 1992). In most plants, the isotopic composition of water does not change due to root absorption and transport through the stem xylem (White et al., 1985). However, more and more studies have shown a difference between the isotope composition of xylem water and plant water sources (Poca et al., 2019), and the fractionation can occur along the root water absorption pathway. This fact does not make the isotope method in the soil-plant-atmosphere useless to track water in the continuum, and  $\delta D$  and  $\delta^{18}O$  can still be used similarly to study water absorption of various plants (Poca et al., 2019).

- Araguás-Araguás, L., Rozanski, K., Gonfifiantini, R., and Louvat, D.: Isotope effects accompanying vacuum extraction of soil water for stable isotope analyses, Journal of Hydrolog, 168, 159–171, https://doi.org/10.1016/0022-1694(94)02636-P, 1995.
- Orlowski, N., Breuer, L., and McDonnell, J. J.: Critical issues with cryogenic extraction of soil water for stable isotope analysis. Ecohydrology, 9(1) 1-5, 2016.
- Sofer, Z., and Gat, J.: The isotope composition of evaporating brines: effect of the isotopic activity ratio in saline solutions. Earth Planet. Sci. Lett. 26 (2), 179–186, 1975.
- Gaj, M., Kaufhold, S., Königer, P., Beyer, M., and Himmelsbach, T.: Mineral mediated isotope fractionation of soil water. Rapid Communications in Mass Spectrometry, 31(3), https://doi.org/10.1002/rcm.7787, 2017.
- Meißner, M., Köhler, M., Schwendenmann, L., Hölscher, D., Dyckmans, J.: Soil water uptake by trees using water stable isotopes (d2H and d18O) a method test regarding soil moisture, texture and carbonate. Plant and Soil 376 (1–2), 327–335, https://doi.org/10.1007/s11104-013-1970-z, 2014.
- Orlowski, N., Breuer, L., Angeli, N., Boeckx, P., and Mcdonnell, J. J.: Inter-laboratory comparison of cryogenic water extraction systems for stable isotope analysis of soil water. Hydrology and Earth System Sciences Discussions, 1-36. https://doi.org/10.5194/hess-2018-128, 2018.

- Walker, G. R., Woods, P. H., and Allison, G. B.: Interlaboratory comparison of methods to determine the stable isotope composition of soil water, Chem. Geol., 111, 297–306, https://doi.org/10.1016/0009-2541(94)90096-5, 1994.
- Walker, G. R., Woods, P. H., and Allison, G. B.: Interlaboratory comparison of methods to determine the stable isotope composition of soil water, Chem. Geol., 111, 297–306, https://doi.org/10.1016/0009-2541(94)90096-5, 1994.
- Poca, M., Coomans, O., Urcelay, C., Zeballos, S. R , and Boeckx, P.: Isotope fractionation during root water uptake by acacia caven is enhanced by arbuscular mycorrhizas. Plant and Soil (3), https://doi.org/10.1007/s11104-019-04139-1, 2019.
- Ingraham, N. L., and Shadel, C.: A comparison of the toluene distillation and vacuum/heat methods for extracting soil water for stable isotopic analysis. Journal of Hydrology, 140(1), 371-387, https://doi.org/10.1016/0022-1694(92)90249-U, 1992.

10. L101: Replace the description with exact data.

Response: We have made changes based on your suggestions, and the revised content is as follows:

The precipitation in the Shiyang River Basin is mainly from July to September, and the average relative humidity in summer (46.79%)and autumn (44.79%)is higher than that in winter(43.82%) and spring(32.18%).

11. L114-115: The purpose of each observing system should be introduced.

Response: We have added the purpose of setting up six observation systems to the manuscript, and the added content is as follows:

Currently, our research using six observing systems includes but is not limited to the following: (1) River source area: We use the stable isotopes to analyze the influence of water vapor source and altitude on precipitation isotope and analyze Hydrological transmission time in arid regions. (2) Oasis area: We use the stable isotope to analyze the impact of human activities on the water cycle. (3) Reservoir system area: We use the stable isotopes to analyze the impact of reservoirs on plant water use strategies and the impact of water conservancy projects on the water cycle in arid regions. (4) Oasis farmland area: We use the stable isotope to analyze the effects of different agricultural models on soil water movement in arid oasis areas. (5) Ecological restoration area: We use stable isotopes to analyze water sources and water use strategies in riparian wetland artificial ecological forests. (6) Salinization area: We use stable isotopes to analyze the evaporation, leakage and storage of soil moisture in different vegetation areas in the lower reaches of the Shiyang River.

12. L128: Information about the device used to collect precipitation, such as pictures, should be added.

Response: We use a rain gauge to collect precipitation. The picture of the rain gauge is shown below:

Rain measuring cylinder and measuring cup

13. L132: This sentence is repeated, and it is recommended to delete it.

Response: We delete this sentence from the manuscript based on your suggestion.

14. L138-139: How to calculate the precipitation isotope value of that day after sampling multiple times of precipitation in one day?

Response: For water samples with multiple precipitations in a day and multiple samplings in a precipitation event, the stable isotope value in the precipitation of that day is the weighted average of the precipitation of all water samples of the day, calculated with the following formula:

$$\overline{\delta X_p} = \frac{\sum P_i \delta X_i}{\sum P_i}$$

Where  $P_i$  is the precipitation of the i-th water sample in a day,  $\delta X_i$  is the stable isotope value of oxygen or hydrogen in the first water sample on that day.

15. L150: Are all soil samples at 10cm intervals? I saw 5cm intervals in the data set. Response: In all soil sampling points, except for the sampling point of DTX, the surface soil is collected in the form of 0-5cm, 5-10cm, and the surface soil of the other soil sampling points is collected in the form of 0-10cm, 10-20cm. Collected in this way in DTX is to analyze the agricultural activities better and crop water use research in the area.

16. L151-152: Are there any replicates for soil samples of each soil layer?

Response: Yes, it is. Two soil samples were collected for each soil layer. Part of the soil samples were dried to measure the soil moisture content, and the other part was used to conduct isotope experiments to obtain soil water isotope values.

17. L155-157: How many plant species are sampled? How about the position of sampled stems in the canopy? What is the size of stem samples? "xylem stem" should be "stem".

Response: In our 12 vegetation sampling sites, we collected a total of 10 vegetation. Among them, the vegetation we collected in BDZ was Agropyron cristatum; the vegetation we collected in CQQ were Corn (leaf), reed, jujube (Branches), dryland willow (Branches); the vegetation we collect in DT is Reed; the vegetation we collect in DTX is Spring wheat (stem), corn (root, stem, leaf); in HJX we collect ear vegetation is Willow (Branches); in HLD The vegetation we collect with HLZ is Qinghai Spruce (Branches); the vegetation we collect in WWPD is Corn (leaf), wheat (leaf); the vegetation we collect in XYWG are Poplar (Branches), wheat; the vegetation we collect in YXB The thing is Corn (leaf); the vegetation we collect in SWX is Corn, wheat (leaf); the vegetation we collect in LLL is Salsola purpurea. Our sampled stems are located at the bottom right of the tree canopy, which means we are collecting the oblique branches of the tree. The sample size of Qinghai spruce and poplar we collected is about 50cm, and the sample size of herbaceous plants such as Salsola is about 10cm. We have changed "xylem stem" to "stem". In addition, we have also added the basic information table of vegetation samples, the newly added table is as follows:

| Sampling points | Vegetation types                                                | Sample size |
|-----------------|-----------------------------------------------------------------|-------------|
| BDZ             | Agropyron cristatum                                             | 30g ±0.5    |
| CQQ             | Corn (stem), reed, jujube (Branches), dryland willow (Branches) | 30g-100g    |
| DT              | Reed                                                            | 30g±0.5     |
| DTX             | Spring wheat (stem), corn (root, stem)                          | 30g±0.5     |
| HJX             | Willow (Branches)                                               | 100g±0.5    |
| HLD             | Qinghai Spruce (Branches)                                       | 100g±0.5    |
| HLZ             | Qinghai Spruce (Branches)                                       | 100g±0.5    |
| WWPD            | Corn (stem), wheat (stem)                                       | 30g±0.5     |
| XYWG            | Poplar (Branches), wheat                                        | 100g±0.5    |
| YXB             | Corn (stem)                                                     | 30g±0.5     |
| SWX             | Corn, wheat (stem)                                              | 30g±0.5     |
| LLL             | Salsola purpurea                                                | 30g±0.5     |

**Table 3 Basic information of plant samples**

18. L141: Which reservoir of water was measured?

Response: Among our sampling points, there are 3 sampling points in our reservoirs, Hongyashan Reservoir(HYSSK), Xiying Reservoir(XYSK) and Nanying Reservoir (NYSK).

19. L142: How is the groundwater sampled? What is the depth of water table at each sampling point.

Response: Groundwater samples were obtained from the groundwater monitoring wells of the Shiyang River Basin Administration, China Hydrological Administration and Gansu Hydrological Administration. The sampling interval is monthly. Groundwater samples were taken from the groundwater monitoring wells of the Shiyang River Basin Administration, the China Hydrological Bureau and the Gansu Provincial Hydrological Bureau. Distributed in the Shiyang River Basin, the sampling interval is monthly. There are few groundwater sampling points in the upper reaches of the Shiyang River. The groundwater depth is between 15m and 30m, the groundwater depth in the middle reaches between 2.5m and 60m, and the groundwater depth in the downstream is between 2.5m and 30m.

20. L145: "telling the date"?

Response: We have revised this sentence in the manuscript, and the revised content is as follows:

Simultaneously, the polyethene bottle sample is labelled with the sample's date, sampling depth, and the stream and tributary stream

21. L146-147: Where is the water sample placed?

Response: We put the collected samples in a refrigerator at about 4° C so that we can perform isotope testing later.

22. L150: What types of soil are collected?

Response: The upper reaches of the Shiyang River Basin are mainly clay, and the middle and lower reaches are clay and sand, but sand is the main soil. Table 2 shows the characteristics of soil in the farmland area of the Shiyang River Basin.

| Soil depth (cm) | Clay (%) | Silt (%) | Sand (%) | Soil bulk density (g/cm 3 ) |
|-----------------|----------|----------|----------|----------------------------------------|
| 0-10            | 10.20    | 38.85    | 50.95    | 1.052                                  |
| 10-20           | 12.94    | 37.76    | 49.30    | 1.194                                  |
| 20-30           | 10.33    | 44.23    | 45.44    | 1.298                                  |
| 30-40           | 13.48    | 38.69    | 47.83    | 1.180                                  |
| 40-50           | 12.01    | 35.09    | 52.90    | 1.140                                  |
| 50-60           | 11.21    | 42.83    | 45.96    | 1.206                                  |
| 60-70           | 10.34    | 42.98    | 46.68    | 1.208                                  |
| 70-80           | 11.09    | 38.96    | 49.95    | 1.106                                  |
| 80-90           | 11.75    | 37.72    | 50.53    | 1.200                                  |
| 90-100          | 7.21     | 35.97    | 56.82    | 1.272                                  |

 Table 2 Basic information of soil samples (Zhu et al., 2021)

Zhu, G. F., Yong, L. L., Zhang, Z. X., Sun, Z. G., Wan, Q. Z., Xu, Y. X., Ma, H. Y., Sang, L. Y., Liu, Yu. W., Wang, L., Zhao, K. K., and Guo, H.W.: Effects of plastic mulch on soil water migration in arid oasis farmland: Evidence of stable isotopes: Catena, doi:10.1016/J.CATENA.2021.105580, 2021.

23. L177-190: The article did not mention the accuracy of the hydrogen and oxygen stable isotope data and the standard samples used in the experiment, which is missing for an article introducing the data.

Response: We have added relevant content to the manuscript, and the added content is as follows:

**4.2.2 Isotope experiment**

All the collected water samples were analyzed in the stable isotope laboratory of Northwest Normal University using liquid water isotope analysis (DLT-100, Los Gatos Research, USA). Each water sample and isotope standard sample were injected six times in a row. We discarded the first two injections and used the average of the last four times as the final result to eliminate the instrument memory effect. The result of the isotope measurement is expressed by the symbol " $\delta$ " and expressed in thousandths of the difference relative to the Vienna Standard Mean Ocean Water (Craig, 1961):

$$\delta_{sample} (\%_0) = \left[ \left( \frac{R_{sample}}{R_{v-smow}} \right) - 1 \right] \times 1000$$
(1)

In the formula, Rsample is the ratio of 18O/16O or 2H/1H in the collected sample, Rv-smow is the ratio of 18O/16O or 2H/1H in the Vienna standard sample.The analytical accuracy of  $\delta^2$ H and  $\delta^{18}$ O are ±0.6‰ and ±0.2‰, respectively.

24. L197: "to test the hydrological data" not "to test the isotopes data".

Response: We have revised this sentence according to your suggestion, and the revised content is as follows:

We use Manner-Kendall to test meteorological and hydrological data, eliminate abnormal values, and use interpolation to obtain vacant values. For the isotope data, we first use the LIMA software to check the original isotope data.

25. L199: How to screen experimental data?

Response: We stipulate that when one or more of the 6 data of a sample is marked in red, we call the sample an error sample, even though the sample is in the display on the liquid water analyzer is normal, and we will re-experiment the sample until it passes the LIMA software verification.

26. L232: Both " $\delta$ D" and " $\delta^2$ H" are used in the manuscript. I suggest use one of them. Response: Thank you very much for your suggestion. We have unified the hydrogen in the article as " $\delta$ D".

27. L233: "...we can found that...".

Response: We have revised this sentence according to your suggestion, and the

revised content is as follows:

In Fig. 3, we can found that.

28. L264-266: This sentence lacks a subject.

Response: We have revised this sentence according to your suggestion, and the revised content is as follows:

According to the precipitation isotope data of the Shiyang River Basin from January 2016 to December 2019 (Figure 5), we used the least square method to obtain the local meteoric water line equation (LMWL):

29. L266-267: Please check the full names of LMWL and GMWL.

Response: We have revised the full names of LMWL and GMWL, and the revised content is as follows:

According to the precipitation isotope data of the Shiyang River Basin from January 2016 to December 2019 (Figure 5), we used the least square method to obtain the local meteoric water line equation (LMWL):  $\delta D = 7.65\delta^{18}O + 9.75$ , compared to the global meteoric water line equation (GMWL),

30. L278-281: What does the data in brackets mean?

Response: "By comparing the slope and intercept of the relation expressions  $\delta^{18}$ O and  $\delta$ D of GMWL and different water bodies, it can be seen that, as far as the slope is concerned, precipitation is the highest (7.65), followed by groundwater (5.11), lake water is the lowest (2.14). There is little difference between the slope of precipitation and groundwater, which means there is a mutual recharge relationship. " The number in parentheses in this sentence represents the slope. The slope of the LMWL is 7.65, the slope of the GWL is 5.11, and the slope of the LWL is 2.14. "In terms of intercept (d), the precipitation was the highest (9.75), followed by the river (-8.44). " The number in parentheses in this sentence indicates the intercept, the intercept of LMWL is 9.75, and the intercept of RWL is -8.44.

31. L309: Both "underground weater" and "ground water" are used in the manuscript.I suggest use one of them.

Response: Thank you very much for your suggestion.We unified the term as "groundwater" in the article.

Figure 1: "Shiyang River system"? Is it "Shiyang River Basin"? Improve the clarity of the picture. The picture in the current manuscript is very blurry, so I can't get relevant information from the picture well.

Response: The Shiyang River system is not the Shiyang River Basin. The Shiyang River system refers to the tributaries and main streams of the Shiyang River Basin. We have improved the clarity of the picture, and the revised picture is as follows: